# Traffic conflict identification method on curved road based on Frenet coordinate system

Jingan Wu[1], Shunying Zhu[1], Ruoxi Jiang ○[2]*, Taotao He[1], Jinquan Chen[1]

**1** School of Transportation and Logistics Engineering, Wuhan University of Technology, Wuhan, China,
**2** Hubei Transportation Investment Group Co., Ltd., Wuhan, China

* 260703@whut.edu.cn

## Abstract

Aiming at the TTC (Time to Collision) and derivative indicators' problems of unclear definition and missed/wrong judgments of traffic conflicts on curved road in the traditional Cartesian coordinate system, a new method that can better identify the conflicts on curved road is proposed. The method first establishes the Frenet coordinate system according to the road centerline (i.e., the reference line), and obtains the vehicle trajectory coordinates in the Frenet coordinate system. The Frenet coordinate system can simplify the calculation difficulty of vehicle trajectory and conflict under the curve road. Then determine the vehicle state in the Frenet coordinate system, and then use TTC to calculate rear-end and lane-change conflicts according to the state of the vehicle (non-lane-change/lane-change). Finally, a total of 4 hours of video data were collected based on the K283 of the lane-switch work zone of the Jiqing Highway. Subsequently, the continuous high-precision conflict data in the region was obtained through the video and conflict identification program, and the traditional method was compared with the new method. The results show that different methods have a significant impact on the identification of the number of serious conflicts. The new method can reduce the missed judgments of serious rear-end conflicts on curved road, especially at the junctions of curved and straight segments (segment 3/4/7/8/9), and can also reduce wrong judgments of serious lane-change conflicts. In addition, among the 125 added serious rear-end conflicts identified by the new method, the maximum deceleration of 10 conflicting vehicles during the conflict exceeds the dangerous state −4/-1.5m/s$^2$, which explain that the new method can help us better identify the risks of curved road. The new method combines the Frenet coordinate system, vehicle state determination and TTC, which can reduce the missed/wrong judgments of conflicts on curved road, and expand the traffic conflict identification from previous straight road to full-line road alignment.

**Data availability statement:** Data cannot be shared publicly because of third-party commercial confidentiality and proprietary information owned by collaborative institutions, which are subject to non-disclosure agreements and institutional regulations. Data are available from the Wuhan University of Technology Data Institutional Data Access / Ethics Committee (contact via email: 2806438371@qq.com) for researchers who meet the criteria for access to confidential data.

**Funding:** This work has received funding from the China National Natural Science Foundation (No. 71771183 to Shunying Zhu) and China National Natural Science Foundation (No. 71901166 to Shunying Zhu).

**Competing interests:** The authors have declared that no competing interests exist.

# 1 Introduction

Various data shows that many countries in the world, especially developing countries, cause a large number of personal casualties and economic losses every year in road traffic accidents. How to reduce the number of accidents and reduce the severity of accidents has always been the most important issue in the field of road traffic safety. Most of the previous traffic safety research is based on historical accident data. Traditionally, most road safety studies have relied on historical accident data. While this approach offers advantages such as logical soundness [1,2], it suffers from several limitations, including difficulties in data collection, small sample sizes of actual crashes, under-reporting of minor incidents, and the inherently reactive nature of accident-based analysis [3,4]. In response to these deficiencies, scholars put forward the concept of traffic conflict in the 1960s and 1970s, and summarized the Traffic Conflict Technique (TCT, Traffic Conflict Technique) [5]. As the previous stage and initial form of traffic accidents, traffic conflict supplements and improves the traffic events model, and has the statistical advantages of "large sample, short period, small area, and high reliability."

In current traffic conflict research, the Cartesian coordinate system is predominantly adopted by default to determine the location of traffic conflict points. On straight road segments, straight-line equations can be used within the Cartesian framework to fit road boundaries, enabling relatively accurate and straightforward estimation and prediction of conflicts based on vehicle trajectory points. However, on curved road segments, due to the diversity of road curvature types, it is difficult to accurately fit road boundaries according to actual conditions. Moreover, when vehicle trajectories span both straight and curved sections, tracking trajectories using road boundaries becomes more complex [6]. The trajectory points of lane-changing vehicles are influenced by both the lane-changing maneuver and the curve equation. Particularly on curved sections where the lengths of inner and outer lanes differ and where the paths of the front and rear wheels deviate, accurately describing traffic conflicts becomes challenging [7]. Consequently, determining trajectory points for lane-changing vehicles on curves under the Cartesian coordinate system is complicated and unsuitable for precisely capturing the dynamic behavior of vehicles on curved segments, which undermines the reliability of conflict identification. Therefore, the accuracy of determining the trajectory point of the lane-change vehicle based on the curve in the Cartesian coordinate system cannot be guaranteed, and the applicability is poor. The Frenet coordinate system is based on the road centerline, which can greatly simplify the calculation of vehicle trajectories and conflicts under the curve road [8].

Additionally, traffic conflict detection techniques rely on conflict indicators for classification. Existing commonly used indicators primarily fall into three categories: Qualitative judgments based on evasive maneuvers (such as turning or significant deceleration) [9–11]; Measures based on spatio-temporal proximity [12,13]; Indicators based on the motion characteristics of traffic entities, such as deceleration rate to avoid conflict (DRAC) [14–16]. Overall, the development of these traffic conflict indicators has significantly contributed to the promotion and application of traffic conflict

techniques as well as to conflict identification. However, most existing indicators are primarily applied to ordinary straight road sections or urban intersections, while studies focusing on curved road segments remain scarce [17–19]. Currently widely used TTC and its derivative metrics (such as TIT, TET [20]), along with DRAC, are all based on a key assumption: "traffic participants maintain their current speed and direction unchanged." Directly applying this assumption to curved sections (using highway transition zones as an example) raises the following issues:

(1) The definition is unclear. In a curve, should "current direction" refer to the vehicle's instantaneous tangential direction (Direction 1) or the actual driving trajectory direction (Direction 2)? Existing research predominantly defaults to Direction 1 (i.e., tangential direction), while Direction 2 better aligns with the vehicle's actual driving path. Although Tarko [21] proposed TD2, which accounts for lateral motion, it has not gained mainstream adoption. (2) The definition is unreasonable. Directly adopting the Direction 1 assumption would lead to two types of misjudgments: Vehicles in the same lane may be erroneously deemed free of rear-end collision risk at bends (underestimation); Lane-change conflict points may be projected outside the road (overestimation), as shown in Fig 1b, which defies common sense.

In recent years, some scholars have addressed the issues of vehicle trajectory tracking and conflict identification on curved roads through various approaches. For example, Li described the motion state of vehicles on curves by combining the curve radius with the degree of steering angle deviation of the vehicle [21]. Hu integrated two different conflict indicators while accounting for their correlation, and applied a bivariate Bayesian hierarchical extreme value modeling method to comprehensively identify traffic risks on curves, thereby reducing the impact of individual indicators on risk assessment [22]. Cai utilized the relative angle between vehicles to accurately identify and warn of hazardous vehicles on both straight and curved sections [23]. A few scholars, such as Tarko [24], have considered two alternative indicators, TD1 and TD2, for curved road segments. However, in most traffic conflict studies, no distinct conflict identification method has been specifically proposed for curved sections; instead, conflict detection on curves is often equated with that on straight segments. Although the aforementioned methods for vehicle trajectory tracking and conflict identification on curves have shown effectiveness in specific scenarios or for particular problems, their underlying logic tends to provide tailored solutions for specific issues. As a result, they often struggle to adapt to varied scenarios and complex situations, and may involve computationally intensive procedures. The Frenet coordinate system, which is constructed based on the road centerline, simplifies the process of "fitting boundaries – tracking trajectories" for vehicle trajectory tracking and exhibits good adaptability to both straight and curved segments. Moreover, identifying traffic conflicts in the Frenet coordinate system allows for the continued use of classic traffic conflict indicators, thereby avoiding the issues of ambiguous and unreasonable conflict definitions mentioned earlier, and ensuring the accuracy and rationality of conflict identification. Additionally, given that traffic conflict techniques generally demonstrate good generalization across multiple scenarios, conflict identification within the Frenet framework also offers strong scenario adaptability. Furthermore, it can significantly reduce the computational complexity involved in vehicle trajectory and conflict analysis on curved sections [25].

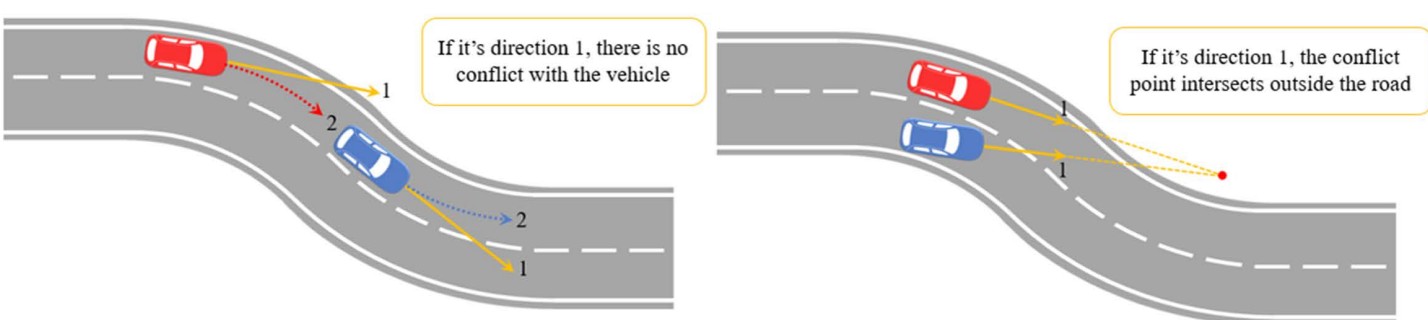

**Fig 1. TTC assumptions are unclear and unreasonable on curved road.**

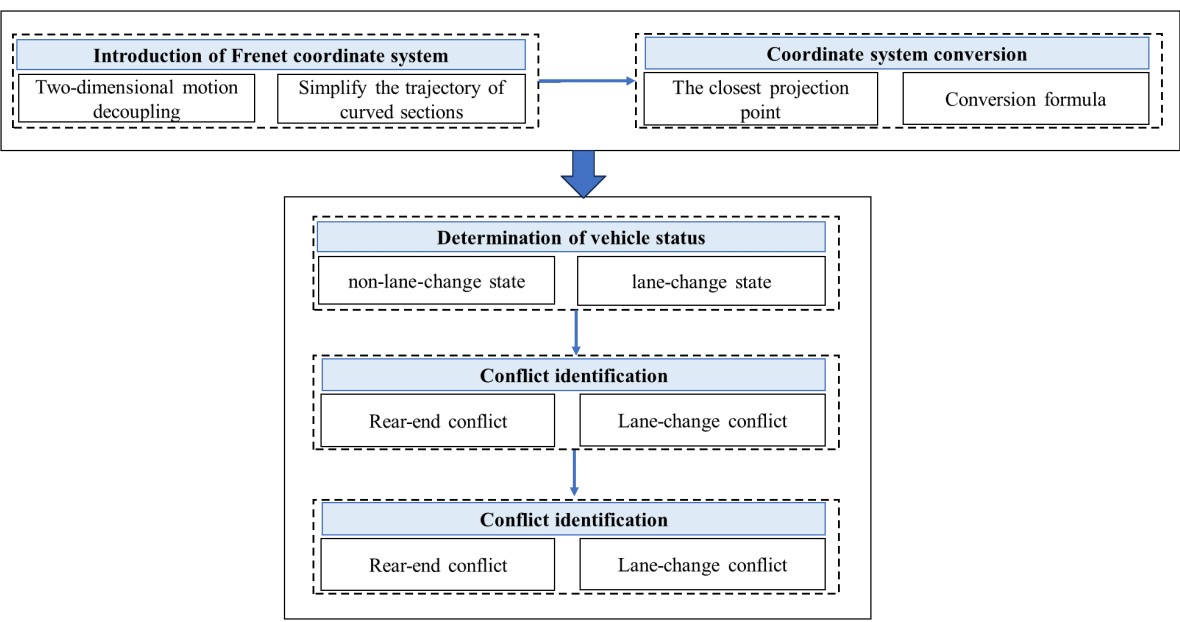 PLOS One

The aforementioned issues indicate that existing Cartesian coordinate systems and conflict indicators exhibit limitations such as inadequate applicability on curved road sections and significant flaws in their underlying assumptions. To address this, this paper proposes a systematic new method for identifying traffic conflicts on curved road sections, starting from the construction of coordinate systems and the definition of conflicts. The method process includes: 1) establishing a Frenet coordinate system according to the road centerline (i.e., reference line). Synchronously convert the vehicle trajectory into the trajectory coordinates in the Frenet coordinate system; 2) Define the determination process under each state of the vehicle (non-lane-change/lane-change) in the Frenet coordinate system; 3) In the Frenet coordinate system, according to the state of the vehicle, Rear-end and lane-change conflicts are calculated based on TTC identification. Then, the two conflict identification methods of "Cartesian coordinate system + TTC" and "Frenet coordinate system + vehicle state determination + TTC" are compared for instance data.

## 2 Methods

This section combines the Frenet coordinate system, vehicle status determination, and TTC indicators to introduce a complete method for implementing full-line (straight line + curve) collision detection using the Frenet coordinate system. The overall architecture diagram is shown in Fig 2.

### 2.1 Introduction of Frenet coordinate system

Traffic conflict identification relies on an appropriate coordinate system. To better identify traffic conflicts on curved road segments, it is essential to first establish a more suitable coordinate system. Typically, the Cartesian coordinate system is used to describe vehicle trajectory positions, as shown in Fig 3a. However, on curved road sections, due to the influence of road curvature, defining potential collision points and travel directions in the Cartesian coordinate system may yield counterintuitive results, making it difficult to accurately describe vehicle trajectories. To address the limitations of the Cartesian coordinate system in conflict identification, this paper adopts a Frenet coordinate system for traffic conflict analysis [26]. In the Frenet coordinate system, the "s"-axis aligns with the road centerline, representing the longitudinal

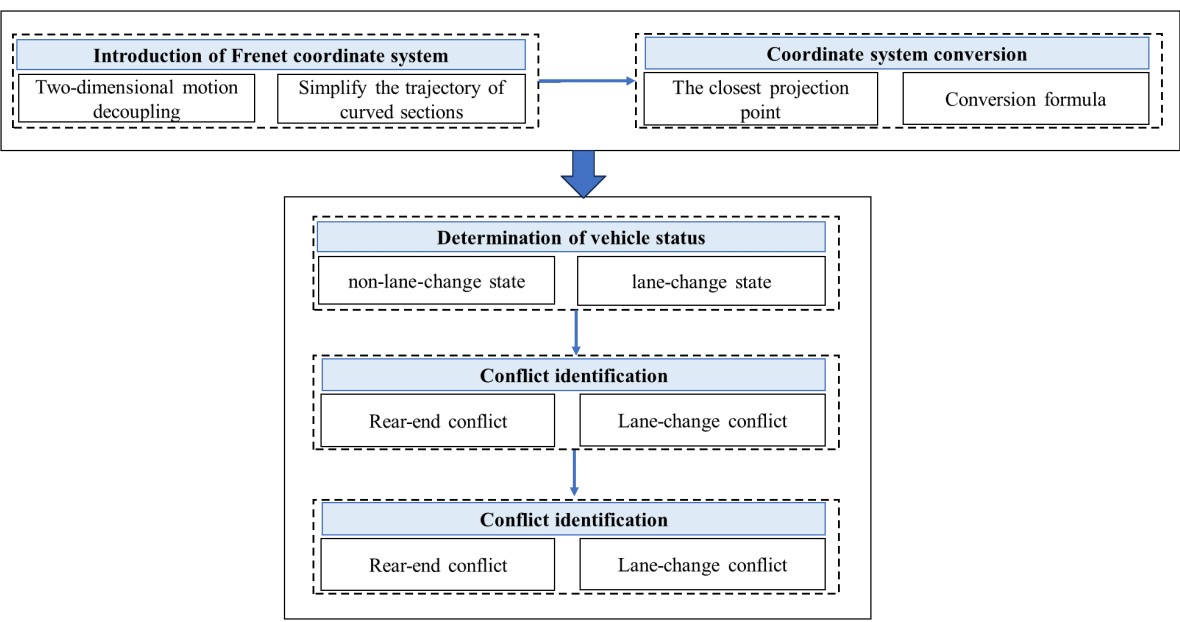

**Fig 2. Method architecture diagram.**

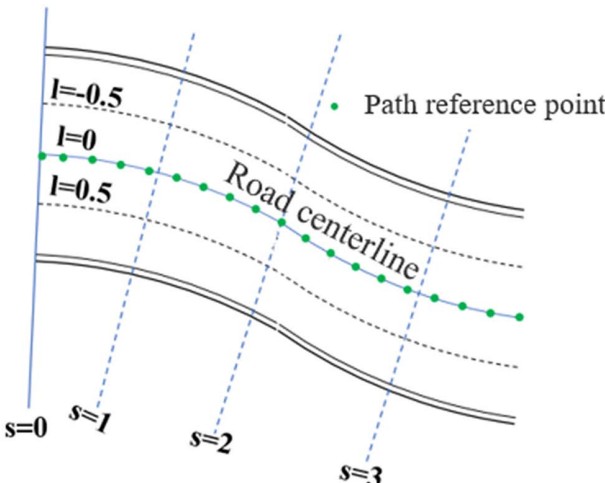

**Fig 3. The direction of the road centerline of the curved road in the Cartesian and the Frenet coordinate system.**

travel distance of the vehicle along the road, while the "l"-axis, perpendicular to the s-axis, represents the lateral displacement distance [27]. The Frenet coordinate system accounts for road curvature, and within this framework, vehicle trajectories are treated as straight lines, as illustrated in Fig 3b. Compared to curved trajectories, straight trajectories are easier to analyze, thereby reducing the complexity of trajectory representation and conflict recognition. Additionally, this paper places reference points along the road centerline at 10-meter intervals and constructs a polynomial-fitted centerline through these points, as shown in Fig 3c. A comparison between Fig 3a and 3b clearly illustrates the differences between the two coordinate systems.

## 2.2 Frenet coordinate system and Cartesian coordinate system conversion formula

In the case of ignoring the height information, the Frenet coordinate system can be simplified as a two-dimensional rectangular coordinate system, that is, only the tangent vector $\vec{t}$ and the normal vector $\vec{n}$ of the driving direction are included. The mapping relationship between the vehicle trajectory and the road centerline in the Cartesian coordinate system and the Frenet coordinate system is shown in Fig 4. The road centerline (reference line) vector is $\vec{r}$, and the current vehicle trajectory point vector is $\vec{p}$. In the Cartesian coordinate system, this vector is generally expressed as $\vec{p} = (x, y)$ (the vertical $z$ coordinate is usually not considered), but in the Frenet coordinate system it is the closest projection point distance $s$ and lateral offset $d$, i.e., $\vec{p} = (s, d)$, are used in the description. Define the azimuth, unit tangent vector and unit normal vector of the current reference line $\vec{r}(s)$ as $\theta_r$, $\vec{T}_r$ and $\vec{N}_r$ respectively, then the azimuth, unit tangent vector and unit normal vector of the current vehicle trajectory point $\vec{x} = \vec{x}(s, l)$ are $\theta_x$, $\vec{T}_x$ and $\vec{N}_x$ respectively. Given a certain point $\vec{x} = [x_x, y_x]$ of the vehicle, obtain $[s, l]$: (1) $s$ Find the projection point $[x_r, y_r]$ closest to $[x_x, y_x]$ on the centerline of the road, and the $s$ at this point is the $s$ of $[x_x, y_x]$ in the Frenet coordinate system. (2) $l$ $\vec{x}$, $\vec{r}$ are the vectors of the vehicle trajectory and the road centerline in the Cartesian coordinate system, respectively, we have $\vec{x} = \vec{r} + l\vec{N}_r$, and the transformation can be:

$$l = (\vec{x} - \vec{r})^T \vec{N}_r = \|\vec{x} - \vec{r}\|_2 \cos\left(\theta_{x-r} - \left(\theta_r + \frac{\pi}{2}\right)\right) = \|\vec{x} - \vec{r}\|_2 \left(\sin(\theta_{x-r})\cos(\theta_r) - \cos(\theta_{x-r})\sin(\theta_r)\right)$$

In the above formula, $\theta_{x-r}$ is the direction angle of the vector $\vec{x} - \vec{r}$, and $\theta_r + \frac{\pi}{2}$ is the direction angle of the unit vector $\vec{N}_r$, then the above formula can be obtained by multiplying the vector by formula $\vec{a} \times \vec{b} = |a||b| \cos\theta$. Here, $\theta$ is the angle between vector $\vec{a}$ and vector $\vec{b}$.

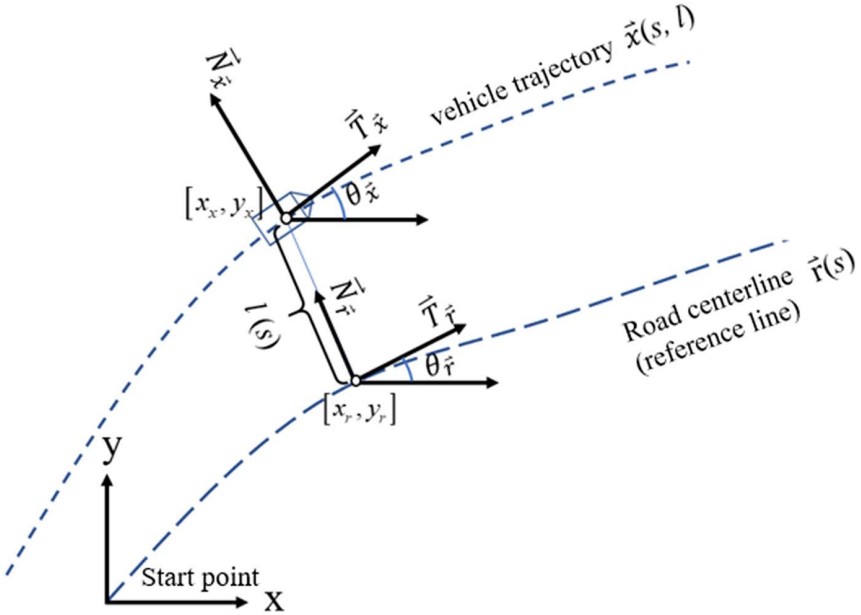

**Fig 4. The mapping relationship between the vehicle trajectory and the road centerline in the Cartesian coordinate system and the Frenet coordinate system.**

Assuming $\vec{x} = (x_x, y_x)$, $\vec{r} = (x_r, y_r)$, then $\|\vec{x} - \vec{r}\|_2 = \sqrt{(x_x - x_r)^2 + (y_x - y_r)^2}$. In the Frenet coordinate system, the vector from each point to the reference line is in the same direction or opposite to the normal vector $\vec{N}_r$ of the reference line, so $\sin(\theta_{x-r}) \cos(\theta_r) - \cos(\theta_{x-r}) \sin(\theta_r) = \pm 1$. $\theta_{x-r}$ is the angle of the vector $\vec{x} - \vec{r}$, therefore,

$$\frac{\sin(\theta_{x-r})}{\cos(\theta_{x-r})} = \frac{y_x - y_r}{x_x - x_r}$$ (1)

and the sign of $l$ can be determined according to the plus or minus of $(y_x - y_r) \cos(\theta_r) - (x_x - x_r) \sin(\theta_r)$. The formula is as follows:

$$l = \text{sign}\left((y_x - y_r) \cos(\theta_r) - (x_x - x_r) \sin(\theta_r)\right) \sqrt{(x_x - x_r)^2 + (y_x - y_r)^2}$$ (2)

According to the above conversion formula, the vehicle trajectory can be converted into coordinate points in the Frenet coordinate system.

## 2.3 Determination of vehicle status

In an ordinary straight road, the conflicting vehicle determines whether it is a rear-end conflict or a lane-change conflict based on the included angle of its current driving direction. However, its definition assumes that there are unclear and unreasonable points on the curved road, and the horizontal and vertical motion of the vehicle on the curved road should be considered, that is, the driving direction of the "2" curve in Fig 1a. Therefore, after obtaining the coordinates of the road centerline and vehicle trajectory in the Frenet coordinate system (**1.2**), we further propose the "vehicle state determination" step. "Vehicle status determination" is divided into two steps: (1) Compare the vehicle trajectory with the road centerline in a small time period in the past, if the vehicle trajectory is parallel to the **road centerline**, the vehicle is determined

to be in a "non-lane-change state"; if the vehicle's trajectory is not parallel to the road centerline, the vehicle is in a "lane-change state." It is worth noting that when the time interval is set too short, misjudgments may occur due to trajectory drift; when the time interval is set too long, the interval may span two states simultaneously, thereby losing the necessity of determining the vehicle's state. This paper adopts a 0.5-second time interval, which can largely avoid the two scenarios mentioned above in practical discrimination while accurately describing the vehicle's lateral and longitudinal motion on curved road sections. (2) In the non-lane-change state, the "current direction" in the assumption becomes "along the direction parallel to the centerline of the road"; In the lane-change state, the "current direction" in the assumption becomes "the vehicle has the direction of travel in the small time period"; In order to highlight the contrast effect, a schematic diagram showing the states of "non-lane-change" and "lane-change" in different coordinate systems is shown in Fig 5. In fact, all steps (1) and (2) should be completed in the Frenet coordinate system.

As can be seen from Fig 5, the "the definition of current direction is unclear" described above has been resolved. The current direction of the vehicle is always related to the road centerline. The vehicle state is judged by the road centerline and the vehicle trajectory. If it is a non-lane-change state, the current direction is parallel to the road centerline; if it is a lane-change state, it intersects the road centerline.

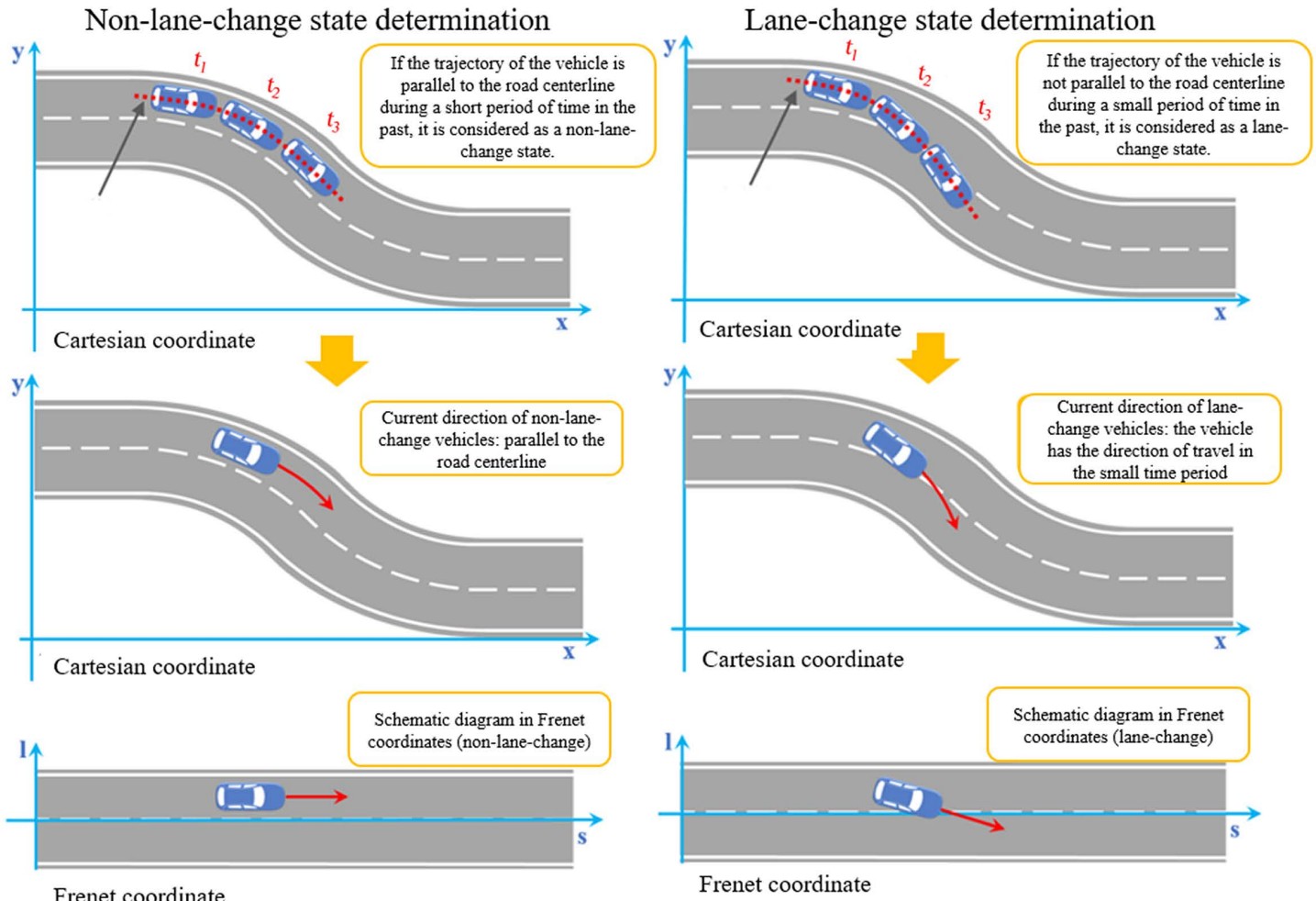

**Fig 5. State (non-lane-change/lane-change) determination process based on the Frenet coordinate system.**

## 2.4 Conflict identification

After completing **1.2** "Coordinate System Conversion" and **1.3** "Vehicle Status Determination," conflict identification can be performed (this paper considers serious conflict identification, see **2.4**). Contrast diagrams of various types of conflict identification based on "Cartesian coordinate system" and "Frenet coordinate system + vehicle state determination" are established respectively: (1) Rear-end conflict. It can be seen from Fig 6a that in the traditional Cartesian coordinate system, although the vehicles 1 and 2 are in the non-lane-change state of following and slack, the TTC under the traditional assumption cannot identify the rear-end conflict. In the Frenet coordinate system and the non-lane-change state determination, the vehicle-following vehicle on a straight and curved road is transformed into a judgment of rear-end conflict on a straight road, so that the rear-end conflict of vehicles 1 and 2 can also be recognized and calculated by TTC (rear-end conflict), reducing missed judgments of rear-end conflicts. (2) Lane-change conflict. It can be seen from Fig 6b that, under the traditional Cartesian coordinate system and traditional assumptions, using TTC (lane-change conflict) identification, the conflict points of the current directions of vehicles 3 and 4 intersect outside the road, which does not conform to common sense. In the Frenet coordinate system and the lane-change state, the lane-change vehicle on a straight and curved road is similar to the judgment of the intersecting vehicle on the straight road, and there are no wrong judgments that the conflict point is outside the road.

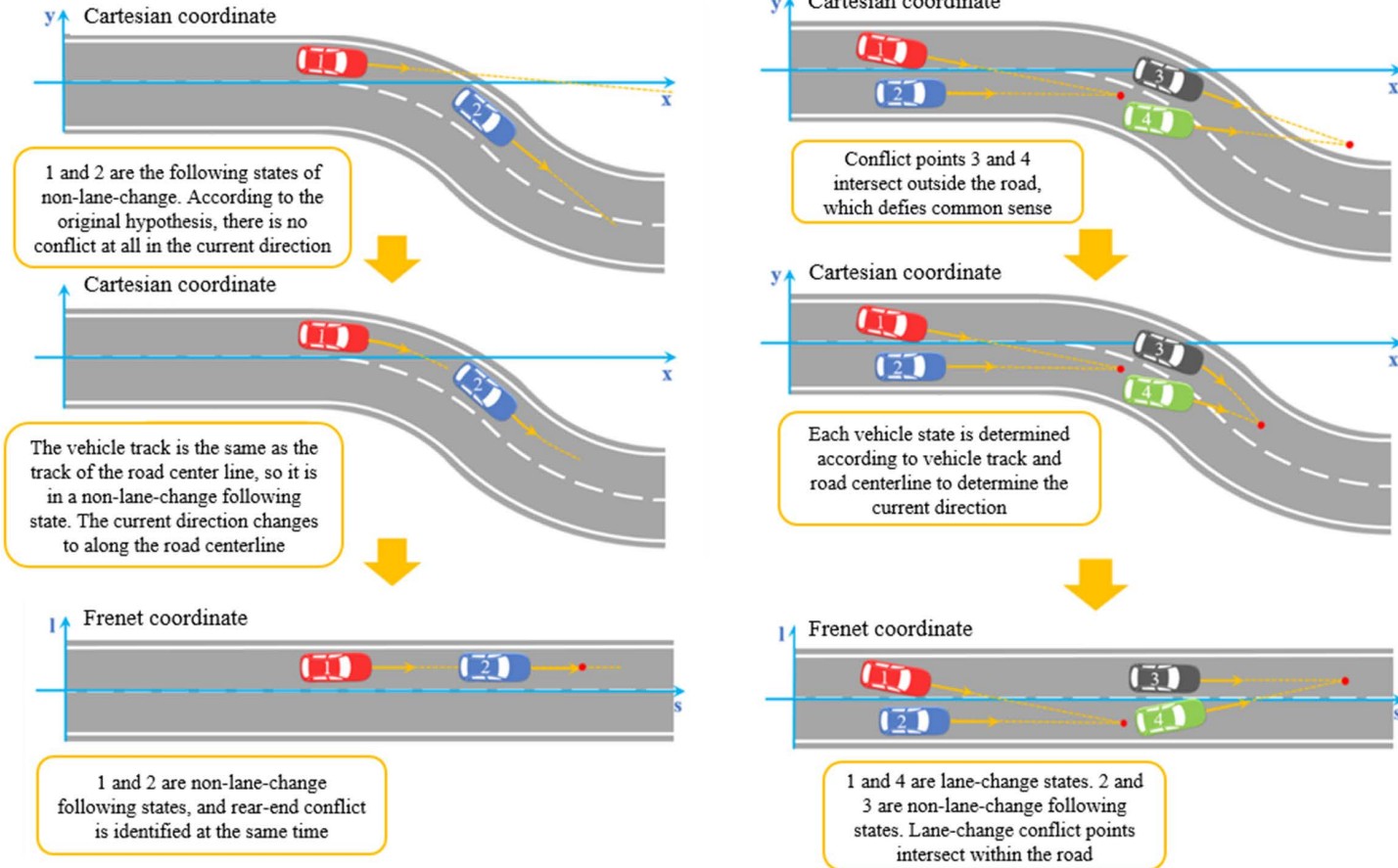

**Fig 6. Comparison of Cartesian and Frenet coordinate system to identify rear-end conflict (a) and lane-change conflict (b).**

## 2.5 TTC indicator calculation formula

This paper uses the TTC indicator to calculate and identify rear-end and lane-change conflicts (this paper only identifies serious conflicts, see **2.4**). Each conflict in the Cartesian coordinate system and the Frenet coordinate system is calculated by the following formula, the difference lies in the difference in the coordinate system and the new "vehicle state determination" (**1.3**) step in the latter. (1) Rear-end conflict identification. (The conflict angle between the front and rear vehicles in the Cartesian and Frenet coordinate systems is 0°.

As shown in <u>Fig 7</u>, the rear vehicle will contact the rear of the front vehicle when the two vehicles collide, and the TTC (rear-end conflict) calculation formula at time $t$ is:

$$TTC_{rear-end} = \frac{S_n - l_{n-1}}{V_n - V_{n-1}}, \quad \forall V_n > V_{n-1}$$

(3)

where:

- $s_n$: The distance between the vehicle $n$ and the head of the vehicle in front (that is, vehicle $n-1$) at time $t$, m;

- $l_{n-1}$: The length of the vehicle $n-1$, m;

- $V_{n-1}$: The instantaneous speed of vehicle $n-1$ at time $t$, m/s;

- $V_n$: The instantaneous speed of vehicle $n$ at time $t$, m/s.

(2) Lane-change conflict identification. (the conflict angle between the front and rear vehicles in the Cartesian and Frenet coordinate systems is 0°–90°) For angular lane-change conflict vehicles, according to this definition, the shape of the vehicle should be considered, and the $x$ and $y$ coordinates should be decomposed before calculation. As shown in <u>Fig 8</u>: the TTC (lane-change conflict) calculation formula at time $t$ is:

$$TTC_{lane-change} = \begin{cases} 0, & \frac{S_n - (l_{n-1} - B_n \cos\theta)}{v_{nx} - v_{(n-1)x}} > \frac{L_{ny}}{v_{ny}} \quad \text{or} \quad \frac{L_{ny}}{v_{ny}} < \frac{S_n + B_n \cos\theta}{v_{nx} - v_{(n-1)x}} \\ \frac{L_{ny}}{v_{ny}}, & \frac{S_n - (l_{n-1} - B_n \cos\theta)}{v_{nx} - v_{(n-1)x}} < \frac{L_{ny}}{v_{ny}} < \frac{S_n + B_n \cos\theta}{v_{nx} - v_{(n-1)x}} \end{cases}$$

(4)

where:

- $S_n$: The distance between the head of the vehicle $n$ and the vehicle $n-1$ in x-axis, m;

- $B_n$: The width of the vehicle $n$, m;

- $L_{ny}$: The distance between vehicle $n$ and vehicle $n-1$ in the y-axis direction, m;

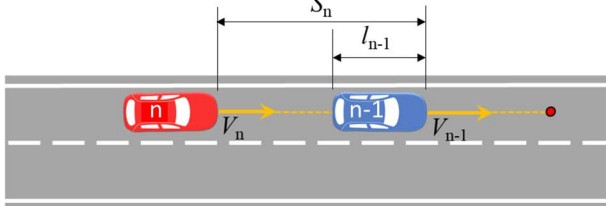

**Fig 7. Calculation of rear-end conflict of TTC indicator.**

<u>https://doi.org/10.1371/journal.pone.0344023.g007</u>

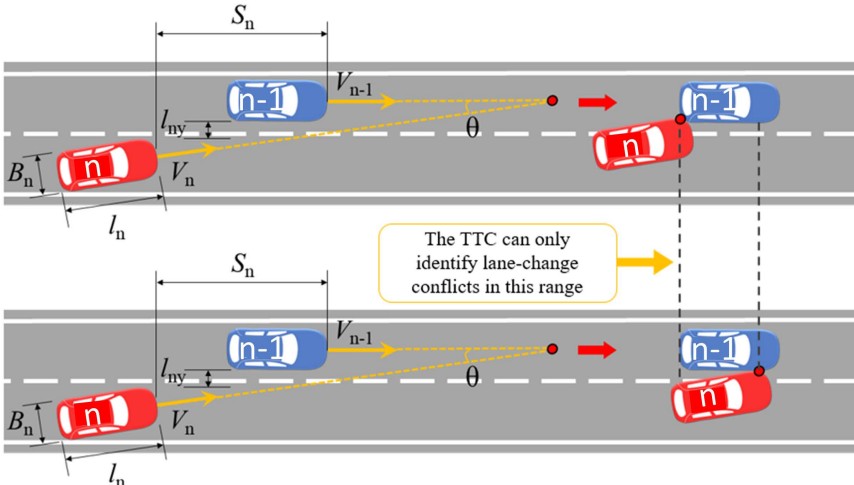

**Fig 8. The traditional TTC indicator lane-change conflict calculation under the Cartesian coordinate system.**

- $v_{nx}$: The instantaneous vehicle speed x-axis component of the vehicle $n$ at time $t$, m/s;

- $v_{(n-1)x}$: The instantaneous vehicle speed x-axis component of the vehicle $n-1$ at time $t$, m/s;

- $v_{ny}$: The instantaneous vehicle speed y-axis component of the vehicle $n$ at time $t$, m/s;

- $\theta$: The angle between the current directions of the two vehicles, °;

## 3 Data

### 3.1 Video capture equipment, location and time

The data collection location is Jinan-Qingdao Highway, Shandong Province, China. This segment features a standard S-curve configuration as well as critical tangent-to-curve transition zones—precisely the core geometric scenarios where traditional methods fail and which this study aims to address. At the same time, the highway work zone, particularly the lane-switch area, provides a realistic and safety-critical operational environment with specific constraints such as speed limits and lane-changing maneuvers, making the conflict analysis more practically meaningful. The layout of the segment enables drone video to capture the entire vehicle trajectory with full, clear coverage, obtaining complete vehicle movement data for straight sections, curved sections, and straight-to-curve transition segments. Therefore, the curved road section studied in this paper takes the lane-changing construction zone of this expressway as an example.

The collection time is from January 2 to January 17, 2018. The collection period includes 4 hours of video data during the morning peak period from 9 to 11 o'clock and the evening peak from 15 to 17 o'clock. The road has 4 lanes in both directions, and the lane width is 3.75m. This paper selects the K283 lane-switch work zone of Jiqing Highway for research. The road lane-switch work zone is two-way and four-lane, with a turning curve road of about 180m long and a speed limit of 60 km/h. During the collection period, the two-way traffic volume was 4534pcu, and the proportion of large vehicles was 21%. The specific collection locations and road segments are shown in Fig 9.

The video capture device uses DJI's PHANTOM 4 PRO drone. The drone has a maximum flying height of 500 meters and a maximum flight time of 30 minutes. The maximum video resolution of the camera video is 4K/60P, and it can be hovered to shoot video, and take GPS positioning. In the experiment, the UAV takes static hovering shooting, the camera is vertically downward, and the flying height is 350-450m. Based on the UAV lens viewing angle parameters, the shooting

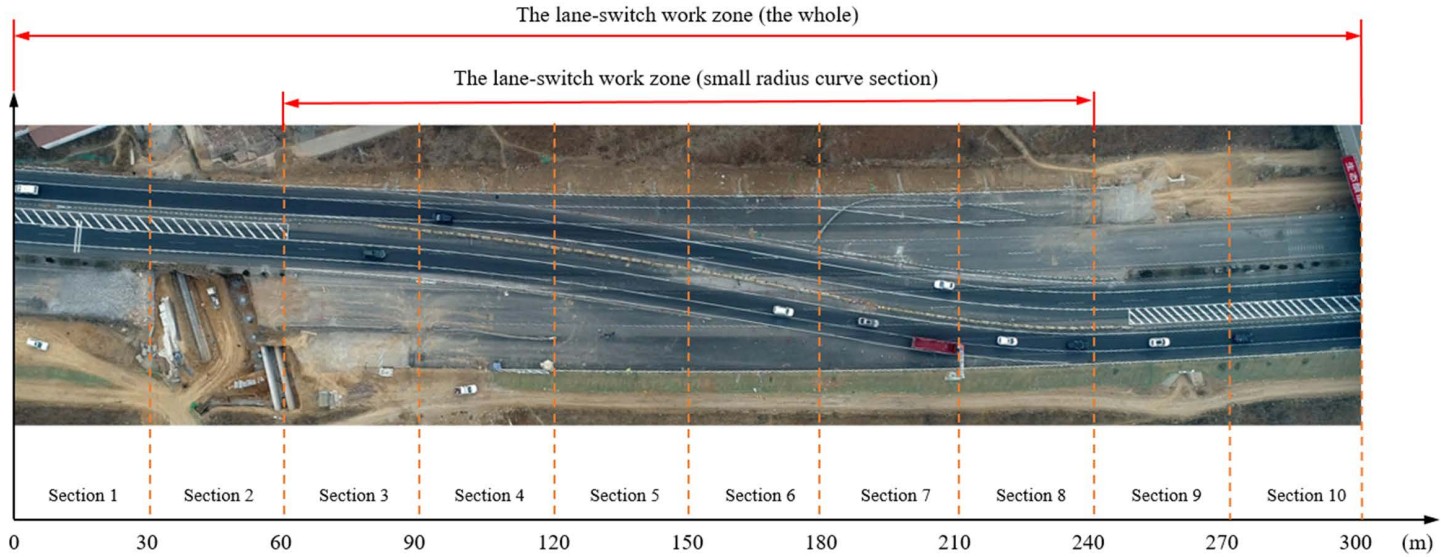

**Fig 9. Jiqing highway K283 road segments.**

range is about 600-700m (length)/300-350m (width). It mainly collects and obtains three major data: 1. Vehicle trajectory coordinate data; 2. Road alignment coordinate data; 3. Traffic conflict data.

### 3.2 Video identification process

After shooting the video, the next step is to perform video identification. The specific process is shown in Fig 10:

(1) Image reading and calibration. Considering the change of high air flow and operation problems, the video shot by the drone has a slight jitter phenomenon, so the following picture will gradually deviate from the original picture. Subsequent frames must be calibrated against the first frame as the reference point to effectively correct the trajectory, as shown in Fig 11.

(2) Vehicle identification. Vehicle identification comprises two components: extraction of the Region of Interest (ROI) and vehicle detection. Given that the Jinan-Qingdao Highway is characterized by a high proportion of large vehicles, high speeds, frequent merging and diverging maneuvers, as well as reduced visibility due to dust, the adjacent frame subtraction algorithm was employed as the method for ROI extraction [28]. For vehicle detection, considering the compatibility of the detection line method with expressway traffic scenarios and its simplicity and efficiency, this approach was adopted for vehicle detection [29].

(3) Vehicle tracking. Various methods exist for vehicle tracking [30]. After evaluating the specific conditions of the Jinan-Qingdao Highway, a spatio-temporal context-based tracking method was selected [31]. This method determines the optimal target position by maximizing the likelihood function of the target location and utilizes fast Fourier transform for learning. Compared to other mainstream approaches, it demonstrates higher accuracy, reliability, and implementation efficiency. The vehicle tracking results are illustrated in Fig 12.

(4) Data output. After completing image calibration, vehicle identification and vehicle tracking, video identification can output the continuous coordinate data of each vehicle every 2 frames (the drone video outputs 30 frames of pictures per second). The format of vehicle continuous trajectory coordinate data is shown in Table 1:

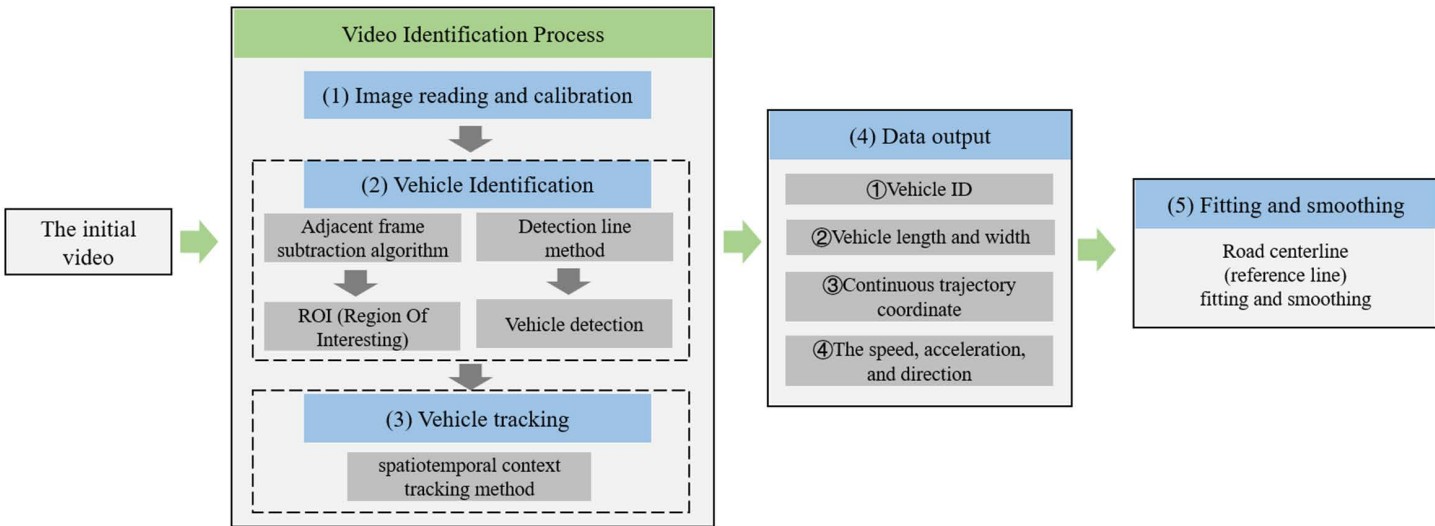

**Fig 10. Video and traffic conflict identification process.**

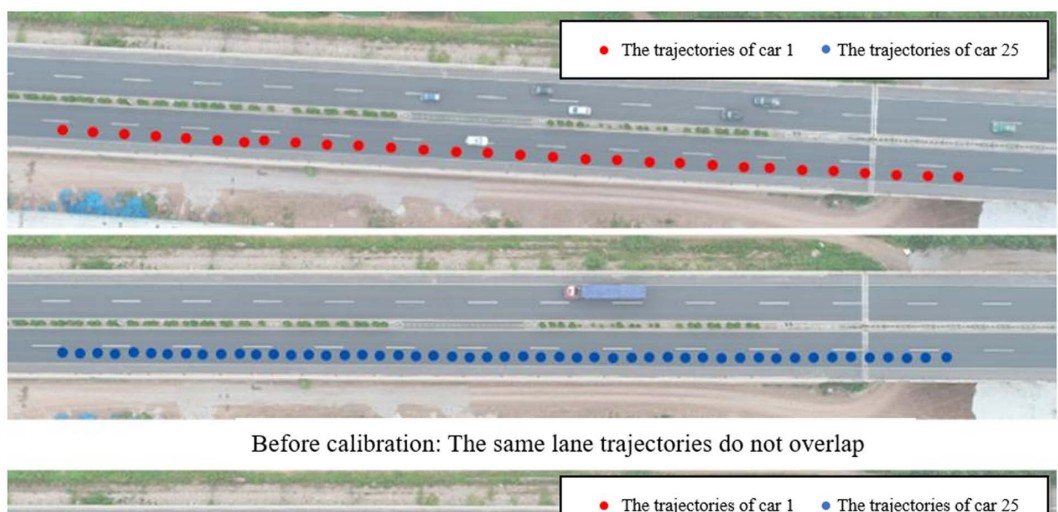

**Fig 11. Comparison of vehicle trajectories before and after calibration.**

According to the continuous trajectory coordinate data, the driving trajectory of each vehicle is drawn, and the schematic diagram of some vehicle trajectories at the collection location is shown in Fig 13. Then, the speed, acceleration, and direction data are obtained according to the coordinates, and the output vehicle motion data type is as shown in Table 2 below.

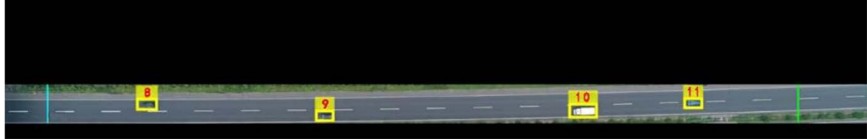

**Fig 12. The vehicle tracking effect.**

**Table 1. Types of vehicle trajectory coordinate data output (part).**

| Frame number | Vehicle ID | X coordinate (m) | Y coordinate (m) | Vehicle length (m) | Vehicle width (m) |
|---|---|---|---|---|---|
| 136 | 12 | 32.5 | 6.0 | 4.4 | 1.8 |
| 136 | 12 | 28.6 | 3.3 | 4.4 | 1.8 |
| 138 | 12 | 34.1 | 6.2 | 4.4 | 1.8 |
| 138 | 13 | 30.1 | 3.3 | 4.4 | 1.8 |

(5) Road centerline (reference line) fitting and smoothing

As mentioned earlier, the road centerline equation is crucial for vehicle state determination and collision detection based on the Frenet coordinate system. Therefore, we need to obtain the road centerline function. The road centerline is modeled by setting path points. If the path points we set on the road centerline are too sparse, converting real-world coordinates to Frenet coordinates will result in angular deviation, which in turn leads to errors in vehicle trajectories. Conversely, setting too many path reference points results in inefficient fitting. After comprehensive evaluation and multiple trials, we determined to place one path reference point every 10 meters along the road centerline. Curves are then fitted through these points to derive the optimal approximation equation for the road centerline, ensuring maximum smoothness (Fig 3b). Path reference point coordinate data can be extracted from video frames using software such as Picpick.

The road centerline function can be fitted in two ways:

① Create a piecewise straight function. For a straight road, the function is established directly based on the two path reference points at the beginning and the end of the straight road. For the non-straight road, it is divided into several short straight lines according to the interval of 10m. Through the idea of limit, multiple short straight lines are used to approximate and fit the non-straight line. The short straight road also uses two consecutive path reference points to establish a function. The final road centerline consists of several piecewise straight functions.

② Spline interpolation. Referred to as Spline interpolation, the polynomial curve function is trained and fitted according to the path reference point coordinates, and then the error function between the predicted point coordinates of each polynomial function and the measured path reference point coordinates is established, and the polynomial function with the smallest error function is selected as the road centerline function. This function can be implemented in the "polynomial trend line" function in EXCEL, and the effect is shown in Fig 14.

Overall, the results of the two methods are close. The former does not have a unified function expression, and the amount of calculation is large. The latter can obtain the polynomial function with the smallest error function without complicated calculation, and has higher precision, so this paper uses the latter for road centerline fitting. When actually fitting the road centerline function, each lane line (or road boundary line) is considered collectively to calibrate the road centerline, thereby reducing errors introduced by the road centerline. Furthermore, considering lane direction variations and differences between lanes in the same direction during observation, multiple lanes in the same direction undergo calibration processing. After error compensation, the final calibrated road centerline is formed.

 

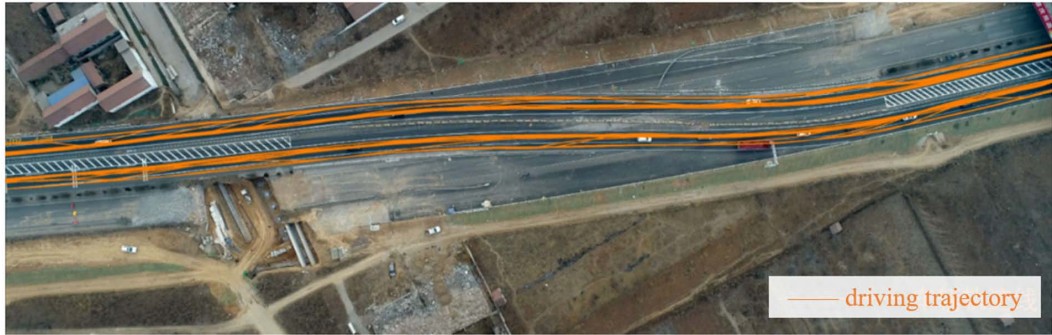

**Fig 13. Schematic diagram of the driving track in the lane-switch work zone (part).**

**Table 2. Types of vehicle trajectory coordinate data output (part).**

| Parameter | Case 1 | Case 2 | Case 3 | Case 4 |
|---|---|---|---|---|
| Frame number | 136 | 136 | 138 | 138 |
| Vehicle ID | 12 | 12 | 12 | 13 |
| X coordinate (m) | 32.5 | 28.6 | 34.1 | 30.1 |
| Y coordinate (m) | 6.0 | 3.3 | 6.2 | 3.3 |
| X speed (m/s) | 4.4 | 4.4 | 4.4 | 4.4 |
| Y speed (m/s) | 1.8 | 1.8 | 1.8 | 1.8 |
| X acceleration (m/s²) | 4.3 | −1.4 | 2.7 | −0.7 |
| Y acceleration (m/s²) | −2.8 | 0 | −2.5 | 0 |
| Vehicle length (m) | 4.4 | 4.4 | 4.4 | 4.4 |
| Vehicle width (m) | 1.8 | 1.8 | 1.8 | 1.8 |

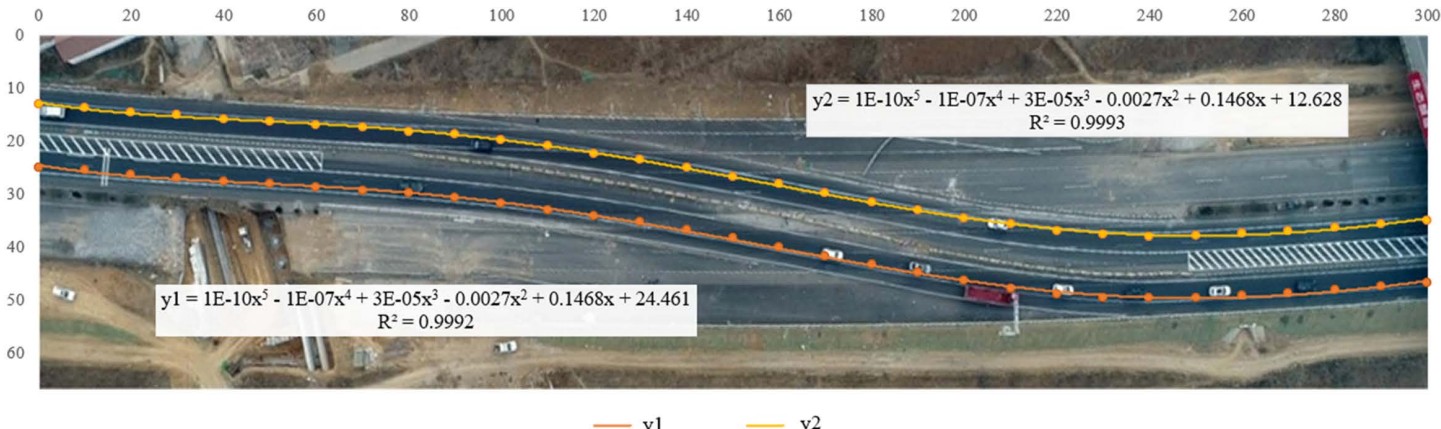

$y2 = 1E-10x^5 - 1E-07x^4 + 3E-05x^3 - 0.0027x^2 + 0.1468x + 12.628$
$R^2 = 0.9993$

$y1 = 1E-10x^5 - 1E-07x^4 + 3E-05x^3 - 0.0027x^2 + 0.1468x + 24.461$
$R^2 = 0.9992$

**Fig 14. Road centerline spline interpolation fitting effect diagram.**

### 3.3 Identification rate and identification accuracy verification

**Identification rate**

Randomly select two videos in a total of about 60 minutes of shooting location K283 for verification. Through statistics, it was found that the video identification software recognized 481 vehicles in total, and continuously tracked 464 vehicles, while a total of 523 vehicles appeared in the video through manual observation. The initial successful identification rate is about 92.0%, and the continuous tracking rate is about 88.7%. The specific data are shown in Table 3:

**Identification accuracy verification**

The roadway dividing line (white dashed line) of all highways in China is 6m long, and the distance between the dashed line and the dashed line is 9m. Therefore, the accuracy and reliability of the video identification program can be judged by this standard.

Randomly select 500 vehicles in some videos, select the coordinate data of each vehicle's random adjacent 2s, and record the x/y-axis displacement of the vehicle within 2s. At the same time, the position of each vehicle in the video is manually marked by Picpick software for comparison. By comparison, it is found that 7.5% of the trajectory errors are below 0.3m, 24.3% of the trajectory errors are below 0.5m, 47.6% of the trajectory errors are below 0.7m, and 84.8% of the trajectory errors are below 1m. Overall, most of the trajectory errors can be controlled within 1m.

### 3.4 Data analysis

**3.4.1 Conflict identification.** For the follow-up comparative analysis, the traditional identification method of "Cartesian coordinate system＋TTC" (hereinafter referred to as "traditional method") and "Frenet coordinate system＋vehicle state determination＋TTC" (hereinafter referred to as "new method") conduct conflict identification separately.

(1) Conflict identification by traditional method.

Under the traditional Cartesian coordinates, the vehicle motion data such as speed, acceleration, distance, direction, etc. are substituted into the definition and formula of TTC (see **1.3** for details) for rear-end conflict and lane-change conflict identification calculation.

(2) Conflict identification by new method. First, the road centerline (reference line) is fitted and smoothed (see **2.2** for details). Convert the vehicle trajectory and the road centerline equation to the Frenet coordinate system (see **1.2** for details). Then carry out the vehicle status determination of the Frenet coordinate system to determine whether it is a non-lane-change or a lane-change state (see **1.3** for details). Finally, according to the TTC definition and calculation formula (see **1.5** for details), Substitute the vehicle motion data such as speed, acceleration, distance, the direction, etc. into rear-end and lane-change conflict identification calculation.

**Table 3. Vehicle identification rate.**

| Parameter | K283 (Test 1) | K283 (Test 2) | Total |
|---|---|---|---|
| Location | K283 | K283 | Total |
| Video Frames | 56610 | 52560 | 109170 |
| Video Duration (s) | 1887 | 1752 | 3639 |
| Initially Recognized Vehicles | 249 | 232 | 481 |
| Continuously Tracking Vehicles | 243 | 221 | 464 |
| Manually Observing Vehicles | 272 | 251 | 523 |
| Initial Identification Rate (%) | 91.5 | 92.4 | 92.0 |
| Continuous Tracking Rate (%) | 92.3 | 88.0 | 88.7 |

In addition, the distribution in terms of nearness to conflicts tells us that serious conflicts are closely related to accidents [32]. According to the previous research results [33], there is no unified conclusion on the TTC serious conflict threshold, and the thresholds are used from 1.0s to 5.0s. Relatively speaking, the TTC severe conflict threshold of 3.0 seconds is more widely applied [34]. Moreover, validating with a larger threshold yields more representative overall results, as it avoids the limitation where the method only holds true under a specific small threshold. Therefore, this paper employs a 3.0-second threshold for rear-end and severe lane-change conflicts to identify severe conflicts, collecting only those with values below this threshold. Minor/general conflicts are excluded from consideration (hereafter, "conflict" and "severe conflict" refer exclusively to severe conflicts below 3.0 seconds).

**3.4.2 Comparison of the number of serious conflicts.** By comparing the number of serious rear-end and lane-change conflicts under the two methods. A total of 4 hours of video data were collected, and a total of 8 groups of video data were divided by the every 30 minutes for statistical analysis.

First, the overall and subsegment (10 segments in total) statistics are carried out, and the paired sample t test is carried out under the two methods. The independent variable is "grouping by different methods," and the dependent variable is "number of conflicts." Significance of the effect on the number of serious conflicts. The main concern is the comparison of methods, testing whether there is a significant difference in the number of conflicts. We are using IBM SPSS Statistics 25 software here.

**3.4.3 Comparison of the ability to identify risks.** Further analysis was made after comparing the number of serious conflicts identified by the two methods. Since the increase in the number of conflicts does not necessarily prove the advantages of the method, further analysis of the relationship between the conflicts identified by the two methods and the real risks is required.

Emergency braking and turning will not only affect the comfort of occupants, but also easily induce traffic accidents [35,36]. Therefore, in this paper, the risk is represented by the maximum deceleration of the x/y-axis during the conflict. After comprehensively referring to some scholars' studies on dangerous driving states [37,38], this paper defines the x/y deceleration thresholds for dangerous state as $-4m/s^2$ and $-1.5m/s^2$ respectively, as long as there is a deceleration on the x/y-axis A value above its threshold is defined as a dangerous state. Such as $-4.1/0 m/s^2$ and $-3 m/-1.6 m/s^2$ are dangerous states. At the same time, as mentioned above, if the rear-end and lane-change TTC exceeds the corresponding threshold of 3.0s, it is defined as a serious conflict.

# 4 Results and discussion

## 4.1 Comparison of the number of serious conflicts

First, the number of serious conflicts identified by the traditional method and the new method is compared. The serious conflict points of K283 are shown in Fig 15 (the red dots are the serious conflict points identified by the traditional method, and the blue dots are the serious conflict points identified by the new method). It can be seen that it is difficult to find the difference between the two methods intuitively. We further divide by segment grouping for detailed statistics.

Then the statistics of serious rear-end and lane-change conflicts under the two methods were carried out by segment, as shown in Fig 15. It can be seen that, except for the straight road segment (segment 1/2/9/10).The numbers of the serious rear-end conflicts identified by the new method are higher than traditional methods (e.g., segment 3: 113−91, segment 4: 105−77, segment 7: 67−50, segment 8: 91−54). It shows that the new method can indeed reduce the missed judgment of serious rear-end conflicts under curved road. The serious lane-change conflicts did not change much. The number of serious lane-change conflicts under the two methods has little change, because the new method does not have much change in the identification of lane-change conflicts compared with the traditional method. The specific reason can be seen in Fig 16. Then, through the paired sample t test, it was found that there were significant differences between the two methods in the number of serious conflicts, especially the rear-end conflicts, as shown in Table 4.

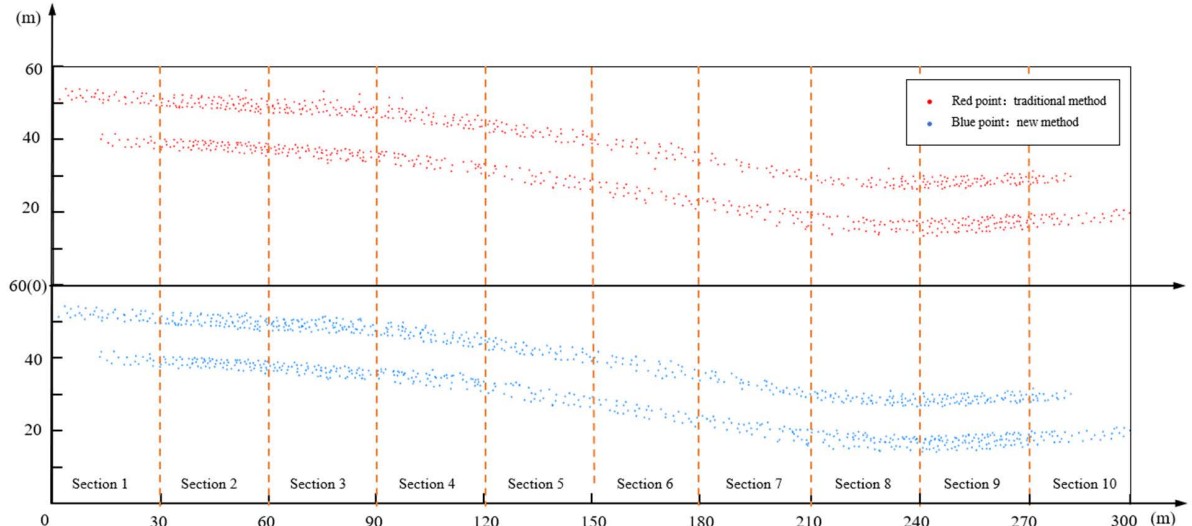

**Fig 15. Serious conflict points identified by traditional and new method.**

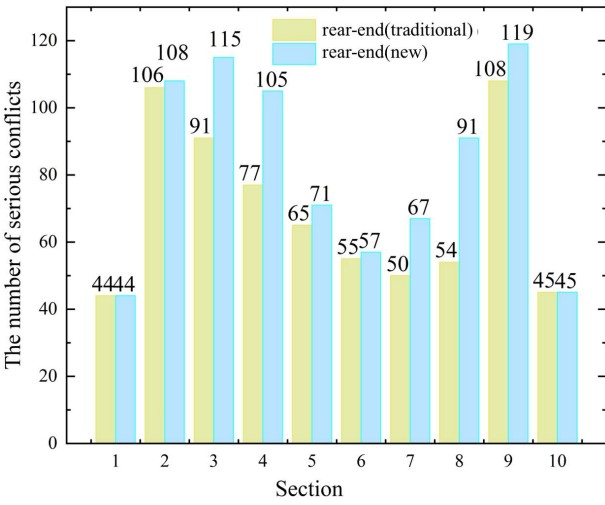

**Fig 16. Comparison of the number of conflicts between the two methods.**

**Table 4. The paired sample t-test results of the two methods.**

| Conflict type | Correlation | Significance | Average value | Standard deviation | T value | Sig.(2-tailed) |
|---|---|---|---|---|---|---|
| *Rear − end* | 0.891 | 0.001 | -12.5 | 13.03 | -3.033 | 0.014 |
| *Lane − change* | 0.998 | 0.000 | -6.0 | 1.17 | -1.616 | 0.14 |
| *Total* | 0.948 | 0.000 | -13.1 | 13.50 | -3.068 | 0.013 |

Table notes Phasellus venenatis, tortor nec vestibulum mattis, massa tortor interdum felis, nec pellentesque metus tortor necnisl. Ut ornare mauris tellus, vel dapibus arcu suscipit sed.

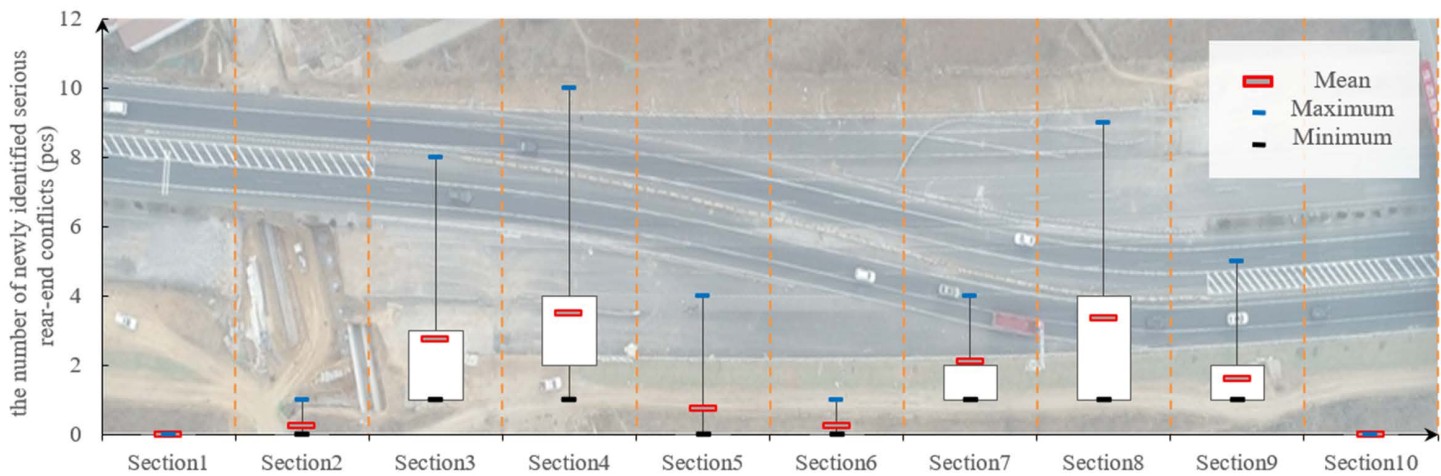

Further we draw a box plot of the data by group and segment (8 groups of 10 segments). Fig 17 shows that, compared with the traditional method, the new method has a larger increase in the number of serious rear-end conflicts in segments 3/4/7/8/9. These segments are often connected to straight segments and curved segments. Purely curved segments and straight segments have little or no increase in the number of serious rear-end conflicts. This is because:

1) The calculation formula of serious rear-end conflict under the two methods is the same (both TTC), the difference lies in the coordinate system and the vehicle state determination. In the straight road segment (segment 1/10), there is no difference between the two coordinate systems, and there is no need to determine the vehicle status, so the two methods identify the same number of serious rear-end conflicts on the straight segment;

2) In the curved road segment (segment 5/6), since in this paper we consider serious conflicts, the distance between the following and leading vehicles is close, so at a relatively close following distance and the low speed of the curved road segment, the line shape of the road segment between the vehicles is similar to a straight line. In addition to the video resolution and identification error, there is little difference between the two methods on the curved road segment;

3) In the connection between the straight road segment and the curved road segment (such as segment 3/4/7/8/9), In the Cartesian coordinate system, a large number of conflict vehicles do not intersect with the current direction. Traditional methods often miss to identify such conflicts. The specific reasons can be seen in Fig 6a. Therefore, the new method has a larger increase in the number of serious rear-end conflicts identified by the traditional method in these segments.

In a lane-change conflict on a curved road, if the direction and speed of the vehicle are assumed to remain unchanged, it may occur that the conflict point intersects off the road, which means that the two vehicles may collide off the road, which is not common sense (See Fig 1b). The new method can overcome this problem. It can be seen from Fig 18 that the traditional method will identify the lane-change conflict on the outside of the road at the curve (red dot). The new method corrected this problem so that the conflict points were all on the road (yellow dots), and a total of 17 serious lane-change conflict points were corrected.

## 4.2  Comparison of the ability to identify risks

Some conflict events are randomly selected, and the TTC values under the two methods are compared, as well as the maximum deceleration of the x/y-axis of the conflicting vehicle during the conflict process. See Table 5 for details.

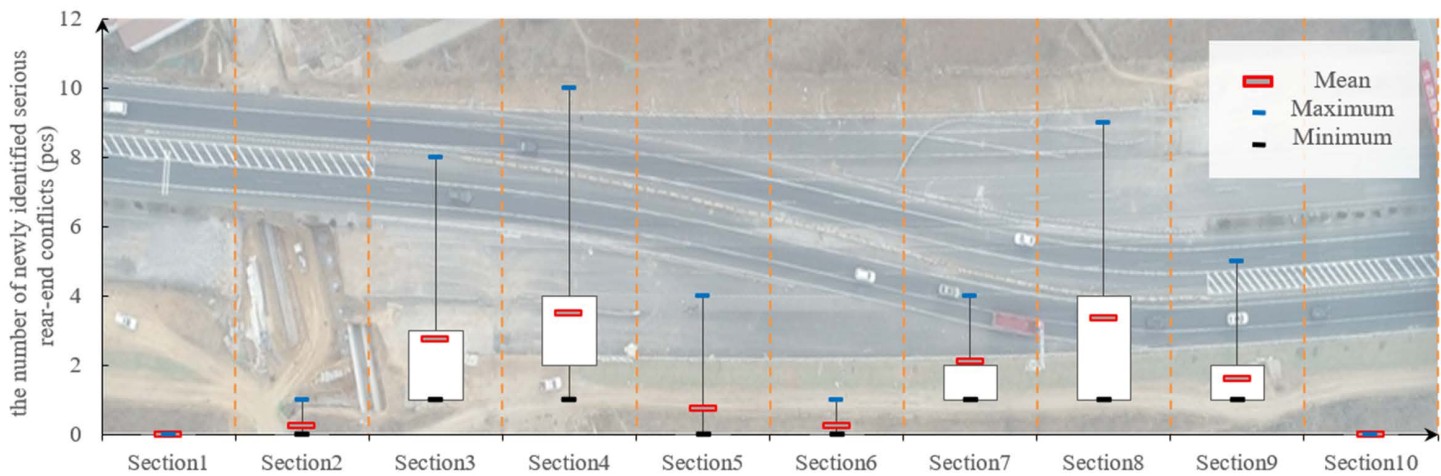

**Fig 17.  A box plot of the new method added to identify serious rear-end conflicts under groups of data.**

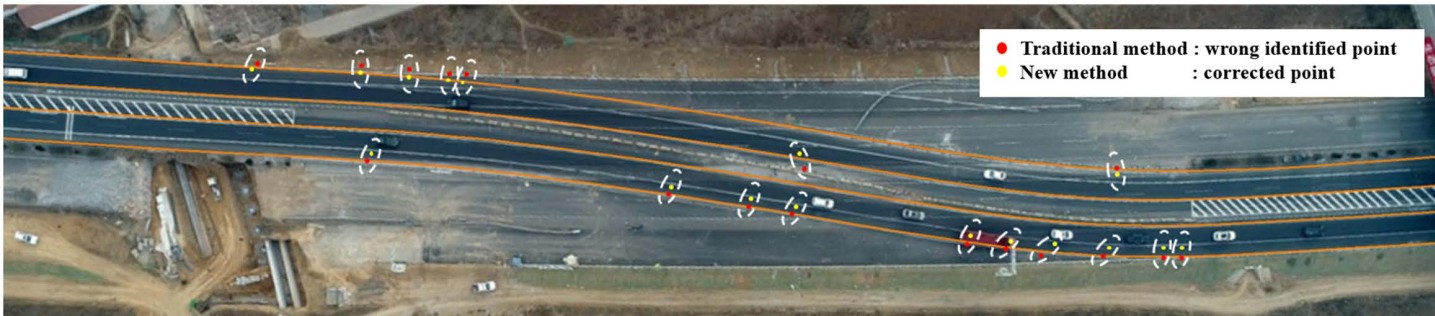

**Fig 18. Comparison of the number of wrong lane-change conflicts identified by the two methods.**

**Table 5. Display of the detailed data of some conflict events.**

| Conflicts ID | Location | Conflict type | TTC(s) of traditional method | Whether traditional method identify | TTC(s) of new method(m) | The maximum deceleration of the x-axis of the conflicting vehicle (m/s2) | The maximum deceleration of the y-axis of the conflicting vehicle (m/s2) |
|---|---|---|---|---|---|---|---|
| 1 | *Segment 1* | *Rear − end* | 1.7 | *Yes* | 1.7 | -4.1 | 0 |
| 2 | *Segment 1* | *Rear − end* | 3.2 | *Yes* | 3.2 | -1.5 | 0 |
| 3 | *Segment 2* | *Lane − change* | 1.9 | *Yes* | 1.9 | -2.8 | -1.2 |
| 4 | *Segment 10* | *Lane − change* | 2.3 | *Yes* | 2.3 | -2.5 | -1.1 |
| 5 | *Segment 3* | *Rear − end* | 2.0 | *Yes* | 2.0 | -2.6 | 0 |
| 6 | *Segment 4* | *Rear − end* | / | *No* | 2.7 | -3.8 | -1.4 |
| 7 | *Segment 7* | *Rear − end* | / | *No* | 1.3 | -5.1 | -1.6 |
| 8 | *Segment 6* | *Lane − change* | 1.6 | *Yes* | 1.6 | -4.2 | -2.4 |
| 9 | *Segment 6* | *Lane − change* | 2.8 | *Yes* | 2.9 | -1.8 | -1.1 |
| 10 | *Segment 7* | *Lane − change* | 2.9 | *Yes* | 2.8 | -1.6 | -0.5 |

From Table 5, it can be seen once again that in the segment 1/2/10 of the ordinary straight road segment, the two methods are almost indistinguishable; at the connection position of the straight road segment and the curved road segment, the number of rear-end conflicts identified by the new method is higher than that of the traditional method. However, the number of conflicts identified by the new method is comparable to that of the traditional method, and the TTC value is slightly different.

Next, the ability of the new method to identify the risks compared with the traditional method is compared. Since the lane-change conflict changes very little in the new method, the following is mainly to analyze the rear-end conflict. First, compared with the traditional method, the new method can identify more serious rear-end conflicts, such as the red dots on the K283 in Fig 19. The maximum x/y-axis deceleration of the conflicting vehicles during these serious conflicts is then plotted as a scatterplot Fig 20. As can be seen from the above, as long as the x/y-axis has a deceleration value exceeding -4m/s²/-1.5m/s², it is a risk. It can be found that among the 125 serious rear-end conflicts identified by the new method, 10 have been in the risk during the conflict, so it can be considered that the new method can identify some new risks. This is also because the new method considers the horizontal and vertical movement of the vehicle on the curved road section, so that conflicts, especially rear-end conflicts, can be more identified, and these conflicts always contain more or less risks.

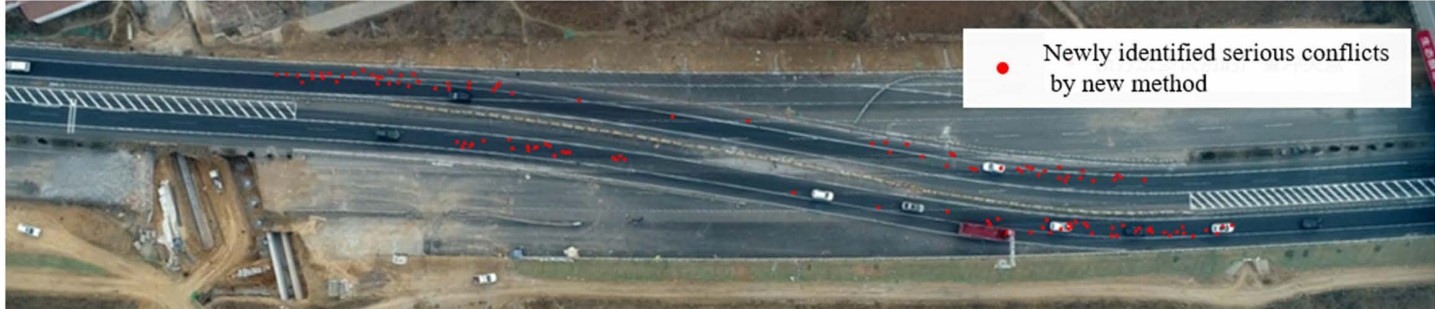

**Fig 19. Serious rear-end conflicts and corresponding decelerations identified by the new method.**

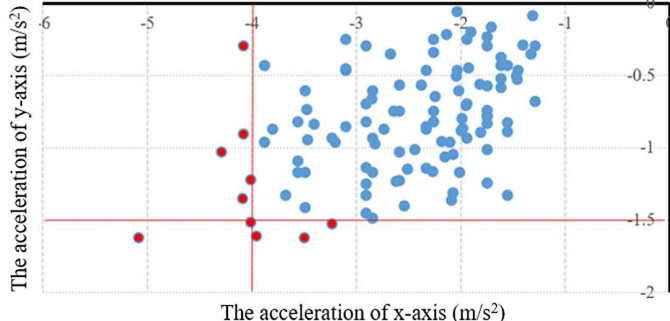

**Fig 20. The more rear-end conflicts identified under the new method with corresponding deceleration and risk.**

## 5 Conclusion

This study successfully addressed the critical limitations of traditional Cartesian coordinate-based traffic conflict measures in curved road segments, where ambiguous definitions and significant misidentification prevail. We proposed and validated a novel, integrated framework for conflict identification specifically designed for curved and transitional sections. The core theoretical contribution of this work is the establishment of a unified analytical framework that systematically combines the Frenet coordinate system with a precise vehicle state (lane-keeping/lane-changing) determination module and the Time-to-Collision (TTC) metric. This framework fundamentally resolves the definitional ambiguity and unreasonable motion assumptions inherent in applying traditional TTC to curves by decoupling vehicle motion into intuitive longitudinal (s) and lateral (l) components relative to the road geometry. It provides a consistent and mathematically rigorous foundation for describing vehicle interactions across both straight and curved alignments, moving beyond the domain-limited approaches of prior research.

The practical implications and actionable insights of this Frenet-based TTC framework are substantial for real-world road safety applications. Firstly, it enables accurate and consistent safety assessment across entire road networks, eliminating the previous methodological discontinuity between straight and curved segments. The framework's ability to correctly map conflict points onto the actual roadway and significantly reduce missed detections (particularly for rear-end conflicts in transition zones) provides transportation engineers and safety analysts with a more reliable tool for proactive hotspot identification, road safety audits, and before-and-after evaluation of geometric design or traffic control measures. Secondly, the output—high-fidelity, georeferenced conflict data—can directly inform the development and calibration of

Advanced Driver Assistance Systems (ADAS) and vehicle-to-infrastructure (V2I) communication protocols for curved sections, enhancing their risk prediction and warning algorithms. The validation results, which showed the detection of previously overlooked dangerous events (e.g., conflicts involving high deceleration rates), confirm the framework's potential to uncover latent risks in existing infrastructure, thereby supporting more targeted and effective safety interventions.

In summary, the proposed framework offers a robust methodological advancement for traffic conflict analysis. While this study is limited by its focus on a specific site and time period, it establishes a validated proof of concept. Future work should expand validation to diverse road geometries, traffic conditions, and integrate with dynamic trajectory prediction models to address the inherent limitations of the Frenet system in high-curvature scenarios. This will further solidify its role as a practical tool for intelligent transportation systems aiming to enhance safety on roads of all configurations.

## Acknowledgments

The data was collected in Jinan–Qingdao Highway in Shandong Province, China

## Author contributions

**Conceptualization:** Ruoxi Jiang, Taotao He.

**Data curation:** Ruoxi Jiang, Taotao He, Jinquan Chen.

**Formal analysis:** Shunying Zhu, Jinquan Chen.

**Funding acquisition:** Shunying Zhu, Jinquan Chen.

**Investigation:** Jingan Wu, Shunying Zhu.

**Methodology:** Jingan Wu.

**Project administration:** Jingan Wu.

**Software:** Taotao He.

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
