## [Decision Letter · Decision Letter 0]

8 Aug 2025

Dear Dr. Jiang,

Thank you for submitting your manuscript to PLOS ONE. After careful consideration, we feel that it has merit but does not fully meet PLOS ONE’s publication criteria as it currently stands. Therefore, we invite you to submit a revised version of the manuscript that addresses the points raised during the review process.

We look forward to receiving your revised manuscript.

Kind regards,

Gen Li

Academic Editor

PLOS ONE

Journal Requirements:

2. Please update your submission to use the PLOS LaTeX template. The template and more information on our requirements for LaTeX submissions can be found at http://journals.plos.org/plosone/s/latex .

4. Please note that PLOS One has specific guidelines on code sharing for submissions in which author-generated code underpins the findings in the manuscript. In these cases, we expect all author-generated code to be made available without restrictions upon publication of the work. Please review our guidelines at https://journals.plos.org/plosone/s/materials-and-software-sharing#loc-sharing-code and ensure that your code is shared in a way that follows best practice and facilitates reproducibility and reuse.

6. Thank you for stating the following financial disclosure:

“This work has received funding from the China National Natural Science Foundation (No 71771183) and China National Natural Science Foundation (No 71901166). The data was collected in Jinan–Qingdao Highway in Shandong Province, China”

7. Thank you for stating the following in the Acknowledgments Section of your manuscript:

“This work has received funding from the China National Natural Science Foundation (No 71771183) and China National Natural Science Foundation (No 71901166). The data was collected in Jinan–Qingdao Highway in Shandong Province, China”

“The author(s) received no specific funding for this work”

8. In the online submission form, you indicated that your data is available only on request from a third party. Please note that your Data Availability Statement is currently missing the name of the third party contact or institution / contact details for the third party, such as an email address or a link to where data requests can be made. Please update your statement with the missing information.

9. PLOS requires an ORCID iD for the corresponding author in Editorial Manager on papers submitted after December 6th, 2016. Please ensure that you have an ORCID iD and that it is validated in Editorial Manager. To do this, go to ‘Update my Information’ (in the upper left-hand corner of the main menu), and click on the Fetch/Validate link next to the ORCID field. This will take you to the ORCID site and allow you to create a new iD or authenticate a pre-existing iD in Editorial Manager.

10. Please ensure that you include a title page within your main document. You should list all authors and all affiliations as per our author instructions and clearly indicate the corresponding author.

11. Please ensure that you refer to Figure 16 and 18 in your text as, if accepted, production will need this reference to link the reader to the figure.

Reviewers' comments:

Reviewer's Responses to Questions

**Comments to the Author**

1. Is the manuscript technically sound, and do the data support the conclusions?

Reviewer #1: Yes

Reviewer #2: Partly

2. Has the statistical analysis been performed appropriately and rigorously?

Reviewer #1: Yes

Reviewer #2: N/A

3. Have the authors made all data underlying the findings in their manuscript fully available?

Reviewer #1: Yes

Reviewer #2: Yes

4. Is the manuscript presented in an intelligible fashion and written in standard English?

Reviewer #1: Yes

Reviewer #2: No

Reviewer #1: The topic is interesting and consistent with the scope of the journal. The authors propose some kind of "generalization" of surrogate measures of safety computation, switching from the Cartesian coordinate system to the Frenet. The latter is well-suited for both straight and curved road segments, whereas the former, as correctly pointed out by the authors, has several shortcomings when dealing with curved segments. The paper provides a fresh point-of-view to the computation of surrogate measures of safety, which is welcome; unfortunately, I believe that the paper could be improved with some additional tests.

1) Line 189, please better motivate the choice of 0.5 s.

2) Lines 352-361. Please clarify if your centerline reference is the painted road centerline (i.e., you are assuming that the painted lines correspond to the actual midpoint axis of the road, which may be questionable). If so, there seems to be a non-negligible deviation from the spline and the painted centerline at certain points (Figure 13), how does this affect your results? Can you add some remarks on this?

3) Lines 397-402. The core of your work is to compare conflicts detected using the "traditional" and the "new" methods. Selecting the threshold to identify conflicts is a crucial operation that could significantly impact the results. Here, however, the TTC threshold is chosen in a rather arbitrary way. The authors could explore more of the literature to justify it, or to test several different thresholds.

4)Figure 15, it could be better to split it into two graphs, one for the rear-end and one for the lane-change.

5) How to verify the superiority of the proposed method by comparing it to the traditional approach?

6) Some literature suggests updating to the latest literature.

Reviewer #2: 1. TTC is commonly used in the field of traffic safety to represent Time to Collision. Modify the Time to Conflict or its abbreviation in this article to avoid confusion. In addition, please unify the expression of “lane hanging” and “lane change” in the manuscript.

2. The current language standard in the manuscript falls below the journal's requirements. There are numerous grammatical errors and logical gaps between paragraphs. Professional editing is recommended. In addition, there are many confusing expressions that are difficult to understand, such as “It is necessary to wait until the accident Analysis can only be vehicleried out after the occurrence, which is ex post facto” in Line 39-40. Also, The numbering (e.g. (1) 1) ①) in this paper is confusing, please unify.

3. The literature review lacks comprehensiveness, particularly regarding studies on traffic conflict identification at curved roads. Furthermore, more recent references should be included.

4. Why choose the Frenet coordination system and what are its typical advantages compared to the Cartesian coordination system?

5. It is suggested to supplement the overall architecture diagram of the proposed method in “1 Method”, presenting the technical framework and logical structure systematically to enhance the clarity of the discussion.

6. Some symbols in the formula are not explained, which increases the difficulty of reading. In Figures 6-7, vehicle labeling and subscript capitalization are inconsistent.

7. In “2 Data”, the data processing part is too lengthy, it is recommended to simplify it. The result analysis part mainly compares the identification performance of the proposed method and the traditional method. How to obtain actual traffic conflicts based on the real data, and how accurate are the above methods compared to the actual conflicts?

8. The contributions of this article are not clear both in the introduction and conclusion sections.

9. Some paragraphs are too long and not easy to read.

**Do you want your identity to be public for this peer review?** For information about this choice, including consent withdrawal, please see our Privacy Policy

Reviewer #1: No

Reviewer #2: No

---

## [Author Response · Author response to Decision Letter 1]

14 Dec 2025

Dear Editors and Reviewers:

Thank you for your letter and for the reviewers’ comments concerning our manuscript entitled “Traffic conflict identification method on curved road based on Frenet coordinate system” (ID: PONE-D-25-40226). Those comments are all valuable and very helpful for revising and improving our paper, as well as the important guiding significance to our research. We have studied comments carefully and have made correction which we hope meet with approval. The revised paragraphs are all highlighted in bold throughout the revised paper. Our response is outlined below, with the reviewer’s comments presented in italics for clarity. Responds to the reviewer’s comments:

Reviewer #1: The topic is interesting and consistent with the scope of the journal. The authors propose some kind of "generalization" of surrogate measures of safety computation, switching from the Cartesian coordinate system to the Frenet. The latter is well-suited for both straight and curved road segments, whereas the former, as correctly pointed out by the authors, has several shortcomings when dealing with curved segments. The paper provides a fresh point-of-view to the computation of surrogate measures of safety, which is welcome; unfortunately, I believe that the paper could be improved with some additional tests.

1.Line 189, please better motivate the choice of 0.5 s.

Thank you for your valuable suggestion. In the original manuscript, the selection of the 0.5s threshold was stated without a detailed explanation of the rationale. Actually, the determination of this threshold involves considerations of two aspects: an overly short time interval may lead to false positives due to trajectory drift, while an excessively long period could cause the segment to span two different states, thereby failing to capture the specific vehicle status. Accordingly, the 0.5s threshold was adopted based on its discriminative performance in practical applications. The justification for selecting this threshold has also been added to the revised manuscript.

It is worth noting that when the time interval is set too short, misjudgments may occur due to trajectory drift; when the time interval is set too long, the interval may span two states simultaneously, thereby losing the necessity of determining the vehicle's state. This paper adopts a 0.5-second time interval, which can largely avoid the two scenarios mentioned above in practical discrimination while accurately describing the vehicle's lateral and longitudinal motion on curved road sections.

2. Lines 352-361. Please clarify if your centerline reference is the painted road centerline (i.e., you are assuming that the painted lines correspond to the actual midpoint axis of the road, which may be questionable). If so, there seems to be a non-negligible deviation from the spline and the painted centerline at certain points (Figure 13), how does this affect your results? Can you add some remarks on this?

Thank you for your valuable comment.The road centerline reference was calibrated based on each lane marking (or road boundary line). Fig. 13 is a schematic diagram. Since the section shown is a transition zone within an expressway reconstruction and expansion work area, the road boundaries appear blurred with shadows, and some of the old markings have not been completely removed in practice. As a result, visual inspection may create the illusion of significant deviation between the fitted spline curve and the actual centerline of the markings at certain locations. This issue was considered during the calibration process of the road centerline and therefore does not affect the results. We have added clarifications regarding the centerline reference in the main text as follows:

When actually fitting the road centerline function, each lane line (or road boundary line) is considered collectively to calibrate the road centerline, thereby reducing errors introduced by the road centerline. Furthermore, considering lane direction variations and differences between lanes in the same direction during observation, multiple lanes in the same direction undergo calibration processing. After error compensation, the final calibrated road centerline is formed.

3. Lines 397-402. The core of your work is to compare conflicts detected using the "traditional" and the "new" methods. Selecting the threshold to identify conflicts is a crucial operation that could significantly impact the results. Here, however, the TTC threshold is chosen in a rather arbitrary way. The authors could explore more of the literature to justify it, or to test several different thresholds.

Thank you for your valuable suggestion. We have substantiated the rationale for threshold selection through literature citations and incorporated this into the main text.

Relatively speaking, the TTC severe conflict threshold of 3.0 seconds is more widely applied[27]. Moreover, validating with a larger threshold yields more representative overall results, as it avoids the limitation where the method only holds true under a specific small threshold. Therefore, this paper employs a 3.0-second threshold for rear-end and severe lane-change conflicts to identify severe conflicts, collecting only those with values below this threshold. Minor/general conflicts are excluded from consideration (hereafter, “conflict” and “severe conflict” refer exclusively to severe conflicts below 3.0 seconds).

4. Figure 15, it could be better to split it into two graphs, one for the rear-end and one for the lane-change.

Thank you for your valuable suggestion. We noted that integrating rear-end collisions and lane-change conflicts into a single graph better reflects the differences in conflict identification. Therefore, we chose to present the conflict count results in one graph. Upon the reviewer's reminder, we realized this approach also makes it difficult to intuitively discern the differences between following conflicts and lane-change conflicts from the graph. Consequently, in the main text, we split Fig. 16 into two sub-graphs, each displaying rear-end conflicts and lane-change conflicts respectively.

(a) rear-end (b) lane-change

Fig. 16 Comparison of the number of conflicts between the two methods

5. How to verify the superiority of the proposed method by comparing it to the traditional approach?

Thank you for your valuable comment. In the results analysis, we demonstrated the superiority of the new method over the traditional method in four aspects.

The new method identified more severe rear-end conflicts than the traditional approach in all non-straight segments, indicating a reduction in missed detections on curves. In contrast, the number of severe lane-changing conflicts remained largely consistent between the two methods. A paired-sample t-test confirmed that the difference in severe conflict counts—especially rear-end conflicts—was statistically significant.

Further analysis using box plots of time- and segment-stratified data (8 groups × 10 segments) revealed that the largest increases in rear-end conflict detection occurred in segments 3, 4, 7, 8, and 9—transition zones between straight and curved sections. Minimal to no increase was observed in purely straight or purely curved segments.

The traditional method misidentified some lane-changing conflicts outside the actual roadway in curved areas. The new method corrected this issue, successfully mapping 17 severe lane-changing conflict points back onto the road.

Among the 125 additional severe rear-end conflicts identified by the new method, 10 involved dangerous vehicle states with deceleration rates exceeding -4 m/s² or -1.5 m/s², confirming that the method can detect previously overlooked risks.

6. Some literature suggests updating to the latest literature.

Thank you for your valuable comment. We have updated some of the references to include more recent literature from the past five years, ensuring the article's relevance to the current research frontier.

Reviewer #2:

1. TTC is commonly used in the field of traffic safety to represent Time to Collision. Modify the Time to Conflict or its abbreviation in this article to avoid confusion. In addition, please unify the expression of “lane hanging” and “lane change” in the manuscript.

Thank you for your valuable suggestion. In the revised draft, we have standardized the corresponding terminology throughout the text, including replacing “Time to Conflict” with “Time to Collision,” “lane changing” with “lane-change,” and replacing “lane switching” with “lane switch” during our comprehensive review.

2. The current language standard in the manuscript falls below the journal's requirements. There are numerous grammatical errors and logical gaps between paragraphs. Professional editing is recommended. In addition, there are many confusing expressions that are difficult to understand, such as “It is necessary to wait until the accident Analysis can only be vehicleried out after the occurrence, which is ex post facto” in Line 39-40. Also, The numbering (e.g. (1) 1) ①) in this paper is confusing, please unify.

Thank you for your valuable suggestion. We have made extensive revisions to the text, enhancing its logical flow and grammatical accuracy to meet the requirements of the target journal. Regarding your comment on lines 39-40, where the phrasing “requires waiting until after an accident occurs to conduct post-event analysis” was unclear, we have revised it to “the reactive nature of accident-based analysis leads to inherent delays in safety intervention” This better conveys the intended meaning. Additionally, following your recommendation, we have reorganized the numbering system throughout the text and standardized its formatting.

3. The literature review lacks comprehensiveness, particularly regarding studies on traffic conflict identification at curved roads. Furthermore, more recent references should be included.

Thank you for your valuable suggestion. We have supplemented the literature review with additional research on traffic conflicts at curves to enhance its comprehensiveness.

Tarko [17] proposed two alternative indicators, TD1 and TD2, for curves. However, most studies still implicitly transfer conflict detection assumptions from straight sections to curved sections without sufficient adjustment.

(1) The definition is unclear. In a curve, should “current direction” refer to the vehicle's instantaneous tangential direction (Direction 1) or the actual driving trajectory direction (Direction 2)? Existing research predominantly defaults to Direction 1 (i.e., tangential direction), while Direction 2 better aligns with the vehicle's actual driving path. Although Tarko [17] proposed TD2, which accounts for lateral motion, it has not gained mainstream adoption.

4. Why choose the Frenet coordination system and what are its typical advantages compared to the Cartesian coordination system?

Thank you for your valuable comment. In the introduction, we outlined two major issues in identifying traffic conflicts using curved segments: ambiguities in definitions and unreasonable definitions. These problems stem from using the Cartesian coordinate system to detect conflicts on curved segments. The Frenet coordinate system offers an alternative perspective on vehicle trajectories by treating curved paths as straight lines, thereby clarifying definitions and simplifying computational processes. In Section 1.1's introduction to the Frenet coordinate system, we briefly explained how the Cartesian coordinate system's definition of spatial points leads to inaccurate conflict identification. We acknowledge that the original phrasing may have been confusing, resulting in an inadequate explanation of this issue. Accordingly, we have revised the relevant text.

Traffic conflict identification relies on an appropriate coordinate system. To better identify traffic conflicts on curved road segments, it is essential to first establish a more suitable coordinate system. Typically, the Cartesian coordinate system is used to describe vehicle trajectory positions, as shown in Fig. 2a. However, on curved road sections, due to the influence of road curvature, defining potential collision points and travel directions in the Cartesian coordinate system may yield counterintuitive results, making it difficult to accurately describe vehicle trajectories. To address the limitations of the Cartesian coordinate system in conflict identification, this paper adopts a Frenet coordinate system for traffic conflict analysis. In the Frenet coordinate system, the "s"-axis aligns with the road centerline, representing the longitudinal travel distance of the vehicle along the road, while the "l"-axis, perpendicular to the s-axis, represents the lateral displacement distance. The Frenet coordinate system accounts for road curvature, and within this framework, vehicle trajectories are treated as straight lines, as illustrated in Fig. 2b. Compared to curved trajectories, straight trajectories are easier to analyze, thereby reducing the complexity of trajectory representation and conflict recognition. Additionally, this paper places reference points along the road centerline at 10-meter intervals and constructs a polynomial-fitted centerline through these points, as shown in Fig. 2c. A comparison between Fig. 2a and Fig. 2b clearly illustrates the differences between the two coordinate systems.

5. It is suggested to supplement the overall architecture diagram of the proposed method in “1 Method”, presenting the technical framework and logical structure systematically to enhance the clarity of the discussion.

Thank you for your valuable suggestion. Following your recommendation, we have added an overall architecture diagram to the “Methodology” section, systematically presenting the technical framework and logical structure to enhance the clarity of our discussion.

This section combines the Frenet coordinate system, vehicle status determination, and TTC indicators to introduce a complete method for implementing full-line (straight line + curve) collision detection using the Frenet coordinate system. The overall architecture diagram is shown below.

Fig. 2 Method architecture diagram

6. Some symbols in the formula are not explained, which increases the difficulty of reading. In Figures 6-7, vehicle labeling and subscript capitalization are inconsistent.

Thank you for your valuable suggestion. We have supplemented the explanations for certain symbols to ensure that all symbols in every formula are fully explained within the text. Additionally, Figures 6 and 7 have been updated to maintain consistent capitalization for vehicle labels and subscripts throughout the figures.

l_{n-1}�The length of the vehicle n-1�m

v_{\left(n-1\right)x}�The instantaneous vehicle speed x-axis component of the vehicle n-1 at time t�m/s

Fig. 7 Calculation of rear-end conflict of TTC indicator

Fig. 8 The traditional TTC indicator lane-change conflict calculation under the Cartesian coordinate system

7. In “2 Data”, the data processing part is too lengthy, it is recommended to simplify it. The result analysis part mainly compares the identification performance of the proposed method and the traditional method. How to obtain actual traffic conflicts based on the real data, and how accurate are the above methods compared to the actual conflicts?

Thank you for your valuable suggestion. We have streamlined the data processing in Section 2 “Data,” simplifying numerous details with weaker relevance. The complete methodology for obtaining actual traffic conflicts from real data is detailed in Section 1 “Methodology.” Additionally, Section 2.4 provides a retrospective overview of the conflict identification processes for both methods. Both methods identify actual conflict data with high accuracy. Typically, in conflict studies, TTC identification serves as the actual conflict outcome, while other methods are used to detect potential omissions or errors, thereby assessing the accuracy of conflict identification. This paper uses the traditional method (TTC identification in the Cartesian coordinate system) as the baseline and demonstrates that the new method achieves improved accuracy in conflict identification res

---

## [Decision Letter · Decision Letter 1]

5 Jan 2026

Dear Dr. JIang,

We look forward to receiving your revised manuscript.

Kind regards,

Gen Li

Academic Editor

PLOS One

Journal Requirements:

Reviewers' comments:

Reviewer's Responses to Questions

**Comments to the Author**

Reviewer #1: All comments have been addressed

Reviewer #3: (No Response)

2. Is the manuscript technically sound, and do the data support the conclusions?

Reviewer #1: Yes

Reviewer #3: Yes

3. Has the statistical analysis been performed appropriately and rigorously?

Reviewer #1: Yes

Reviewer #3: Yes

4. Have the authors made all data underlying the findings in their manuscript fully available?

Reviewer #1: Yes

Reviewer #3: (No Response)

5. Is the manuscript presented in an intelligible fashion and written in standard English?

Reviewer #1: Yes

Reviewer #3: Yes

Reviewer #1: The authors have addressed all the concerns raised in the previous round, and the manuscript has been notably strengthened in terms of parameter justification, methodological validation, and literature support. A few minor suggestions are as follows:

1. The introduction and literature review could be further enriched by including key references that specifically outline research progress and existing methodological gaps in conflict identification on curved road sections.

2. The theoretical contributions (e.g., proposing a unified framework for curved road conflict identification) and practical implications (e.g., supporting safety assessments in intelligent transportation systems) of this study should be summarized more explicitly.

3. The manuscript would benefit from additional language polishing.

Reviewer #3: This paper proposes a traffic conflict detection method based on the Frenet coordinate system, aiming to address the identification bias of the conventional Time-to-Collision (TTC) metric on curved road segments caused by misalignment between the coordinate system and actual vehicle motion. The research topic is of clear practical relevance: with the increasing availability of high-precision trajectory data, accurately assessing microscopic traffic safety risks under complex road geometries has become a critical challenge in intelligent transportation systems and proactive safety applications. By adopting the Frenet coordinate system, the authors effectively decouple longitudinal and lateral vehicle dynamics on curves and demonstrate—using real-world trajectory data—the advantages of their approach in detecting severe conflict events such as rear-end collisions. Notably, the proposed method exhibits superior risk-capturing capability compared to traditional Cartesian-based methods, especially in transition zones between straight and curved segments.

The manuscript is generally well-structured and reflects a conscientious response to the reviewers’ previous comments. Terminology and language expression have been noticeably improved, particularly through substantial enhancements in the methodological framework diagram, symbol definitions, and consistency across figures and tables, all of which contribute to better readability.

Nevertheless, there remain several areas where the paper could be further strengthened. The specific modification suggestions are as follows:

1. Introduction needs to be further revised to highlight the existing research gaps.

The current introduction provides a general overview of traffic safety and conflict detection but fails to clearly articulate the specific limitations of existing Time-to-Collision (TTC) methodologies in curved road environments. While it mentions the reactive nature of crash-based analysis, it does not sufficiently foreground the core methodological gap—namely, that conventional TTC frameworks rooted in Cartesian coordinates are ill-suited for accurately capturing vehicle dynamics on curves due to misalignment between coordinate axes and actual travel direction.

2. Why are minor accident cases not utilized? It is recommended to provide supplementary explanations.

The introduction points out that one of the limitations of the traditional accident-based data method is that "minor accidents or severe conflicts that do not lead to accidents are often not recorded, yet they contain a wealth of safety information." However, the main text only focuses on "severe conflicts" with TTC < 3.0s, completely excluding consideration of minor/general conflicts. It fails to illustrate whether the new method is capable of capturing such unrecorded events, nor does it explain how to use such information to supplement safety assessments, which inadequately connects with the original intention of "supplementing the traffic event model" mentioned in the introduction.

3. Additional explanations regarding the rationale for selecting the research road segment are required to strengthen the rationality of the sample.

The current study has not provided any elaboration on the rationale for the selection of the core research road segment. It only directly mentions that data was collected from a specific segment without explaining why this segment is suitable as a case for research on traffic conflicts on curved roads. Such an omission results in a lack of support for the scientificity and rationality of sample selection, leaving readers unable to judge whether the characteristics of this segment can represent the general attributes of the target research objects (e.g., curved roads), thereby affecting the credibility and promotional value of the research conclusions.

4. Literature review needs enhanced focus and integration of recent advances.

The literature review currently presents relevant studies in a somewhat fragmented manner and lacks critical synthesis, particularly regarding recent developments in trajectory-based conflict detection (post-2020). Although Tarko’s alternative indicators (TD1/TD2) are mentioned, the discussion does not adequately contextualize how these relate to—or differ from—the proposed Frenet-TTC framework.

5. Conclusions need greater specificity on practical implications and limitations.

The conclusion section remains overly descriptive and does not sufficiently differentiate this study from prior work in terms of actionable insights. It should explicitly state how the Frenet-based TTC framework can inform real-world applications

This paper has a complete framework, clear research logic, and conclusions with certain theoretical and practical value. However, it is necessary to improve methodological details and the depth of discussion, supplement explanations on minor accident cases and the rationale for research road segment selection. It is recommended to review after major revision.

**Do you want your identity to be public for this peer review?** For information about this choice, including consent withdrawal, please see our Privacy Policy

Reviewer #1: No

Reviewer #3: No

---

## [Author Response · Author response to Decision Letter 2]

8 Feb 2026

All specific responses to reviewers and editors can be found in the "Response to Reviewers" document.

---

## [Editor Report · Decision Letter 2]

15 Feb 2026

Traffic conflict identification method on curved road based on Frenet coordinate system

PONE-D-25-40226R2

Dear Dr. Jiang,

We’re pleased to inform you that your manuscript has been judged scientifically suitable for publication and will be formally accepted for publication once it meets all outstanding technical requirements.

Kind regards,

Gen Li

Academic Editor

PLOS One
---

## [Editor Report · Acceptance letter]

PONE-D-25-40226R2

PLOS One

Dear Dr. Jiang,

I'm pleased to inform you that your manuscript has been deemed suitable for publication in PLOS One. Congratulations! Your manuscript is now being handed over to our production team.

Kind regards,

on behalf of

Dr. Gen Li

Academic Editor

PLOS One